# Metabolic modeling elucidates phenformin and atpenin A5 as broad-spectrum antiviral drugs against RNA viruses

Alina Renz [1,14], Mirjam Hohner[2,3,14], Raphaël Jami[2,3], Maximilian Breitenbach[2,3], Jonathan Josephs-Spaulding[4,13], Johanna Dürrwald[2], Lena Best [4], Victoria Dulière[5,6,7], Chloé Mialon[5,6,7], Stefanie M. Bader[8], Georgios Marinos [4], Nantia Leonidou [1,3,9,10], Filipe Cabreiro [11], Marc Pellegrini [8], Marcel Doerflinger[8], Manuel Rosa-Calatrava[5,6,7], Andrés Pizzorno [5,6,7], Andreas Dräger [1,3,12,15] ✉, Michael Schindler [2,3,15] ✉ & Christoph Kaleta [4,15] ✉

The SARS-CoV-2 pandemic has reemphasized the urgent need for broad-spectrum antiviral therapies. We developed a computational workflow using scRNA-Seq data to assess cellular metabolism during viral infection. With this workflow we predicted the capacity of cells to sustain SARS-CoV-2 virion production in patients and found a tissue-wide induction of metabolic pathways that support viral replication. Expanding our analysis to influenza A and dengue viruses, we identified metabolic targets and inhibitors for potential broad-spectrum antiviral treatment. These targets were highly enriched for known interaction partners of all analyzed viruses. Indeed, phenformin, an NADH:ubiquinone oxidoreductase inhibitor, suppressed SARS-CoV-2 and dengue virus replication. Atpenin A5, blocking succinate dehydrogenase, inhibited SARS-CoV-2, dengue virus, respiratory syncytial virus, and influenza A virus with high selectivity indices. In vivo, phenformin showed antiviral activity against SARS-CoV-2 in a Syrian hamster model. Our work establishes host metabolism as druggable for broad-spectrum antiviral strategies, providing invaluable tools for pandemic preparedness.

From December 2019 on, the outbreak of the novel Severe Acute Respiratory Syndrome Coronavirus 2 (SARS-CoV-2) caused a pandemic with dramatic health-related and socioeconomic consequences. This novel virus was highly similar to SARS-CoV, responsible for a global outbreak in 2002 and 2003[1]. In 2012, another coronavirus spread in the Middle East, causing the Middle East Respiratory Syndrome, leading to 2458 reported cases and a high mortality rate of 35%[2]. History shows that pandemics have repeatedly plagued humankind. In the last 100 years alone, there have been four major influenza pandemics and multiple epidemics, including the Spanish flu in 1918, with an estimated 17–50 million deaths worldwide[3], as well as the Asian and Hong Kong flu in 1957/58 and 1968/69 with one to four million deaths worldwide[4]. Pandemics are not only virus-driven: One of the most extensive pandemics was the Black Death from 1331 to 1353, caused by the bacterium *Yersinia pestis* and is estimated to have killed approximately half of Europe's population[5].

Thus, pandemics and epidemics are recurrent, and more are likely to follow in the future, in particular, due to human impact on the global environment[6,7]. Moreover, the COVID-19 pandemic illustrates its substantial impact on long-term socioeconomic well-being due to the widespread and long-lasting consequences of efforts of pathogen containment[8] and the potential occurrence of post-acute sequelae such as long COVID[9]. Therefore, rapidly developing effective treatment strategies and vaccines are vital to mitigate those consequences. However, despite the unprecedented acceleration in the development of treatment and vaccination approaches against SARS-CoV-2, it still took ~10 months before vaccine approval to treat SARS-CoV-2 and another 6–8 months before production and distribution workflows became functional for a widespread roll-out. Thus, treatment approaches that contain viral replication of not only a single but a broad array of viruses as first-line therapeutic approaches for novel

A full list of affiliations appears at the end of the paper. ✉e-mail: andreas.draeger@informatik.uni-halle.de; michael.schindler@med.uni-tuebingen.de; c.kaleta@iem.uni-kiel.de

emerging pathogens are a highly sought-after goal in preparation for future pandemics[10]. Nowadays, *pandemic preparedness* summarizes efforts to provide such broadly acting antivirals.

Due to the essential dependence of viruses on the metabolic networks of their host for reproduction[11], the utilization of in silico models of virally infected cells provides new avenues to identify druggable targets for antiviral therapy[12]. One such approach is represented by constraint-based modeling[13] with flux balance analysis in particular (FBA,[14]) which allows us to predict the metabolic behavior of biological systems. These methods build upon genome-scale metabolic networks that encompass the entire known repertoire of metabolic reactions taking place in an organism[15]. In the context of viral replication, constraint-based modeling allows the simulation and prediction of the viral capacity to replicate within a host cell by considering nutrient availability, energy resources, and other cellular factors[16,17]. Thus, genome-scale metabolic models of host cells extended to incorporate viral replication[16] can be employed to identify host enzymes essential for viral replication but dispensable for cellular viability. The viral biomass reaction is a key element in these networks that represents the sum of all the cellular components and resources consumed or created during viral replication[16]. Through simulation of the viral biomass reaction, important factors influencing viral replication can be identified, such as essential enzymes or the availability of specific nutrients[17]. Importantly, these methods also allow for the integration of OMICs data such as transcriptomics, proteomics, and metabolomics to reconstruct metabolic models that more accurately reflect the metabolic state of cells in a given condition, so-called context-specific metabolic models[18].

In this work, we introduce a computational workflow that we developed to integrate a generic metabolic model of a virally infected cell with transcriptomic data to predict metabolic pathways relevant for viral replication. We demonstrate the establishment of this workflow and how it is exploited to rapidly identify druggable targets and human approved compounds with antiviral efficacy in vitro and in vivo.

## Results

### Development of a computational workflow to predict viral replication capacities and antiviral targets

We developed a computational workflow to reconstruct the metabolic state of virally infected cells, leveraging single-cell sequencing gene expression data (Methods). To this end, we expanded the generic human metabolic reconstruction Recon 2.2[19] by incorporating reactions specific to the substrates essential for the replication of the investigated viruses. Additionally, we considered all metabolites known to exist in the blood as potential inflow to the model (see "Methods"). Subsequently, single-cell sequencing data, were preprocessed with StanDep[20] to identify core sets of reactions active in each cell which then were used to identify context-specific metabolic networks that contain those reactions using fastcore[21]. This process facilitated the generation of context-specific models for each cell or gene expression dataset. The resulting computational models of virally infected cells were then employed to predict the cellular capacity to produce virions via flux balance analysis[14] and to screen cells for enzymes whose knockout impedes viral replication. Moreover, these models enabled us to predict the effect of knockouts on cellular viability and thereby subsetting the target enzymes to those whose inhibition hinders viral replication but not normal cellular metabolism. These predicted targets were further integrated with additional experimental information on the relevance of enzymes for viral replication to subselect candidates for experimental testing.

### SARS-CoV-2 infection systemically activates metabolic pathways to enhance cellular viral replication capacity

In the first step, we used scRNA-Seq data of samples from COVID-19 patients as input to our modeling workflow to predict viral replication capacity depending on cell type and disease severity[22]. Prior experimental observations reveal that viral replication heavily relies on profound changes in host metabolism[23]. We found that the predicted capacity to sustain viral replication in the upper respiratory tract of infected individuals was enormously increased compared to uninfected participants (Fig. 1A). During infection, both ciliated and secretory cells showed a mean increase in the predicted viral replication capacity by a factor of 2 and 3, respectively. Both cell types are the primary site of cellular infection in the upper respiratory tract[24,25]. Also, FOXN4 cells, only detected in infected individuals[22], showed a similar high viral replication capacity. Notably, these changes were not attributable to active viral replication since most cells in the dataset were negative for SARS-CoV-2 RNA. This indicates that viral infection of a particular cell might have pleiotropic effects on non-infected bystander cells, making them more permissive to viral replication. Accordingly, our models predicted a strong induction of several metabolic pathways in non-infected cells of COVID-19 patients based on an increased number of reactions active in the respective context-specific metabolic models, which was even more pronounced in patients with severe disease (Fig. 1B). Thus, 55 of the 57 analyzed metabolic pathways were significantly induced in infected individuals compared to healthy controls, and 39 were significantly more active in patients with severe versus moderate disease. Altogether, this indicates that viral replication heavily depends on a substantial induction of metabolism and supports the notion that inhibition of host metabolism might be used as an antiviral strategy.

### Metabolic enzymes essential for viral replication are enriched among the interactome of human-pathogenic viruses

In the next step, we used our models to identify potential metabolic targets to inhibit viral replication. Besides SARS-CoV-2, we included influenza A and dengue as two highly human pathogenic prototypic RNA viruses to identify promising broad-spectrum antiviral targets. For the identification of druggable targets, we considered three classes of enzymes: "tier-1 targets", "tier-2 targets", and "other enzymes." Tier-1 targets correspond to enzymes whose knockout impedes viral replication while having a minimal impact on the simulated cellular viability. In contrast, the knockout of tier-2 targets inhibits viral replication and impacts normal cellular metabolism. While it is, in principle, not advisable to impede normal cellular metabolism, antiviral treatments are typically expected to be provided shortly during the acute phase of infection. Hence, tier-2 targets follow a similar strategy to chemotherapeutics in cancer, which often impair regular cellular metabolic activity. Finally, enzymes not belonging to either group are categorized as "other enzymes." We separated enzymes into these three classes for each dataset and virus using flux balance analysis (see "Methods"). We hypothesized that the proteome of the viruses should preferentially interact with enzymes relevant to their replication without harming cellular viability and hence, be considered tier-1 targets. In line with this assumption, for all three viruses, we observed a highly significant enrichment of known physical interaction partners of the viral proteome among predicted tier-1 targets compared to tier-2 targets and all other enzymes (Fig. 2).

### Identification of drug targets for broad-spectrum antiviral therapy

Following our identification of tier-1 targets that inhibit the individual viruses, we sought to identify potential broad-spectrum antiviral targets. To this end, we collected all predicted tier-1 targets across all cells from the individual datasets and identified 254 enzymes that occurred as shared tier-1 targets in at least one cell across all datasets (Supplementary Data 6 and Supplementary Data 1). Using the BioGRID database[26] as a reference, we find that interactions of 158 of these targets with human pathogenic viruses have been reported before, which represents a highly significant enrichment among reported interactions between enzymes and viruses (Fisher's exact test $p = 9.1 \times 10^{-12}$, odds ratio 1.9–3.4) and supports their central role in viral replication across diverse viruses. Performing an enrichment analysis, we identified several metabolic subsystems in which shared predicted tier-1 targets were particularly prevalent (Fig. 3A). We observed the most concentrated enrichment among oxidative phosphorylation enzymes, the citric acid cycle, and fatty acid oxidation. This aligns with the high energy need associated with viral replication[27] and the mitochondria's central role in viral replication and antiviral immune responses[28]. Thus, viruses often target

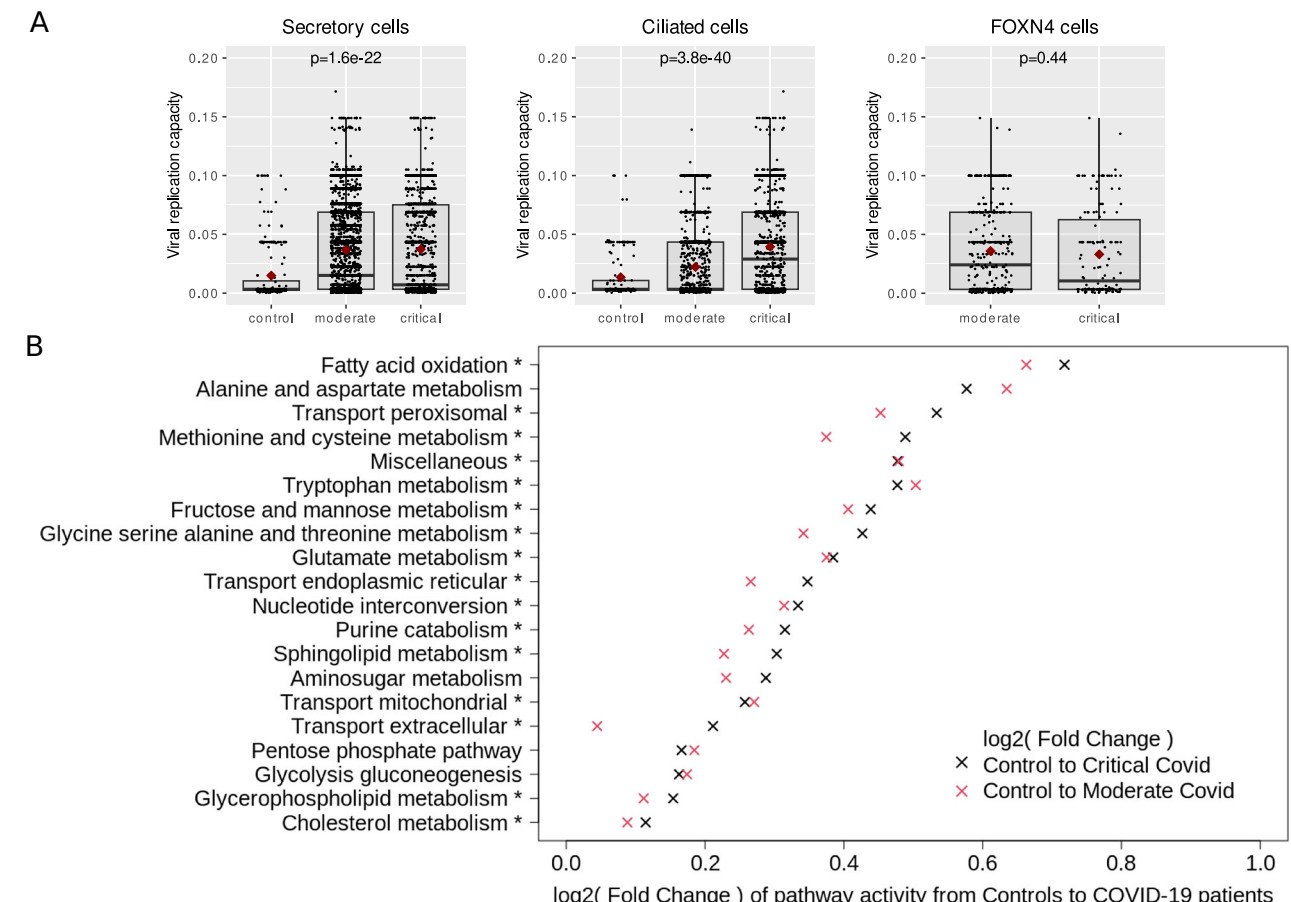

**Fig. 1 | Workflow and impact of disease state on predicted viral replication capacity of SARS-CoV-2 during infection. A** Differences in viral replication capacity according to disease severity in SARS-CoV-2 permissive lung cell types. Please note that FOXN4-positive cells were only detected in infected individuals. *P* values indicate significance of difference of viral replication capacities between groups based on a Kruskal-Wallis test. **B** Cellular metabolism is strongly induced in the respiratory tract of COVID-19 patients. Metabolic models built from throat swabs and lung lavage scRNA-Seq data from SARS-CoV-2 infected individuals were analyzed. Only cells where no viral RNA was detected were considered (="uninfected cells"). Pathways with significantly different predicted activity between moderate to severe COVID-19 patients based on a Wilcoxon test are marked with an asterisk. The 20 pathways with the most pronounced effects are shown. For the complete list of pathways, see Supplementary Data 4.

mitochondrial proteins to deregulate cellular metabolism, providing substrates for viral replication and hampering immune response[29]. Indeed, almost all metabolic reactions in the TCA cycle and the respiratory chain are predicted as targets (Fig. 3B). Along those lines, mitochondrial transport reactions that provide substrates for the TCA cycle are enriched among the predicted targets. However, it is important to note that the enrichment of predicted targets among enzymes known to interact with viruses is still significant when removing enzymes of the respiratory chain and TCA cycle (Fisher's exact test $p = 9.3 \times 10^{-4}$, odds ratio 1.3–2.6).

Interestingly, the knockout of the respiratory chain is predicted to affect viral replication mostly but, to a lesser extent, normal biomass production. Exploring this observation in more detail using the BALF-2 dataset (data from patients with COVID-19), we find that among the ~140k sequenced cells, enzymes of complex I of the respiratory chain were identified in 12.8% of the cells as tier-1 target and in 3.6% of the cells as tier-2 target. Thus, complex I is also essential for maximal biomass production in some cells, but viral replication appears to be much more dependent on it. This characteristic is also reflected in comparing the ATP maintenance costs of the normal biomass reaction of the human model and the viral replication reaction. While the production of one gram of cellular biomass requires 5.9 mmol of ATP, the production of 1 g of virions requires 19.6 mmol of ATP, supporting a much higher energy demand of viral replication.

Other enriched subsystems are N-glycan synthesis as well as degradation and glycolysis/gluconeogenesis. While protein glycosylation is paramount for enveloped viruses[30], we did not include glycosylated proteins in the viral biomass reaction due to a lack of information on virus-specific protein glycosylation. Instead, tracing the corresponding metabolic pathways, we found that those pathways were used to recycle sugars contained in glycans such as mannose, fucose, and glucosamine for use as substrates for viral replication (Supplementary Data 2). Similarly, glycolysis is often found to be induced during viral infection since it also provides crucial building blocks for viral replication[11,31].

To further stratify our hitlist to potential broad-spectrum antiviral targets, we required that each predicted target had an experimentally confirmed interaction with at least two of the three viruses analyzed. Thus, we obtained a list of 39 predicted targets belonging to 22 proteins/protein complexes (Fig. 3B, Supplementary Data 8). Please note that using protein interactions as a selection criterion alone would yield 84 targets (Supplementary Data 5). Among those 22 protein complexes, we selected four, encompassing 11 of the 39 enzyme targets, for further experimental validation based upon additional criteria such as evidence from CRISPR-Cas9 screens[32–34], relevance in other viral diseases, and the availability of inhibitors, as discussed below. These four protein complexes are the NADH:ubiquinone oxidoreductase complex (complex I of the respiratory chain), the succinate dehydrogenase complex (complex II of the respiratory chain), CDP-diacylglycerol—inositol 3-phosphatidyltransferase (CDIPT), and solute carrier family 16 member 1 (SLC16A1). We evaluated the sensitivity of our selected targets to variations in the cut-off thresholds for maximal

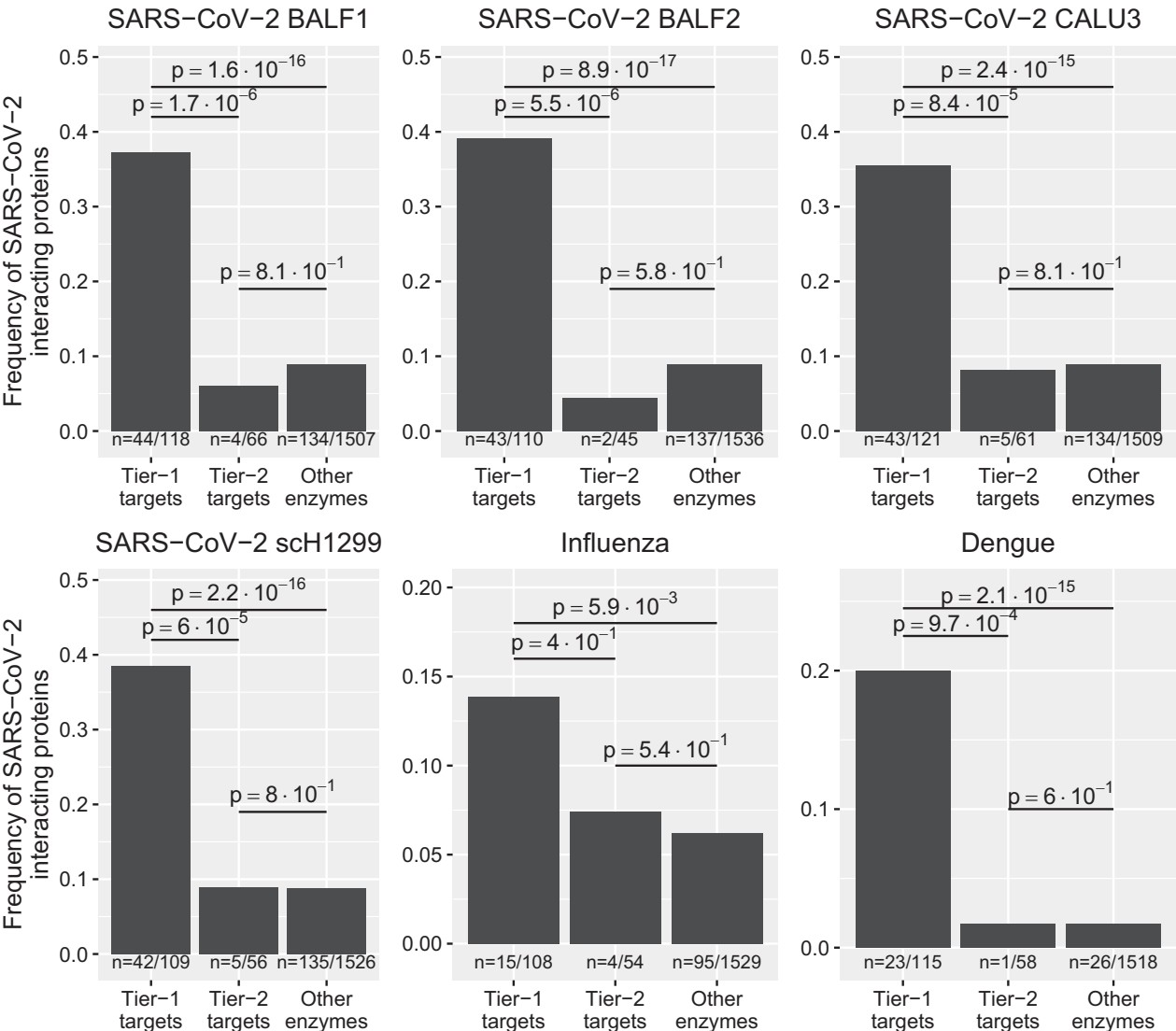

**Fig. 2 | Enrichment of viral interaction partners among model-predicted enzymes involved in the replication of SARS-CoV-2, dengue virus, and influenza A virus.** The fraction of experimentally identified viral proteome interaction partners among tier-1 and tier-2 targets (found in at least 5% of cells with tier-1 targets) was calculated for each dataset. Enrichment was tested using a one-sided Fisher's exact test. *P* values were corrected for multiple testing using false discovery rate control. The number of interacting proteins relative to the number of all proteins in each category is shown below each bar. For the host-virus-interaction data set and the list of tier-1 and tier-2 targets for each data set, see Supplementary Data 5 and 6. ScRNA-Seq datasets: SARS-CoV-2, BALF1[118]; SARS-CoV-2, BALF2[22]; SARS-CoV-2, CALU-3 cell culture[119]; SARS-CoV-2, scH1299 cell culture[119]; influenza H1N1[88], Human airway epithelia (MucilAirTM)[120]; dengue, PBMC,[87]. Protein-protein-interaction datasets: SARS-CoV-2[49]; dengue virus[101]; influenza A virus[102].

viral replication rate and maximal biomass production rate. Our analysis revealed that targets associated with complex I, complex II, and SLC16A1 were largely insensitive to changes in these cut-offs. In contrast, CDIPT was identified only at cut-off values of 80% or higher for maximal biomass production rate, but not at more stringent thresholds (Fig. 3C).

Eight targets correspond to various subunits of NADH:ubiquinone oxidoreductase, for which protein-protein interactions with all three viruses have been observed (Supplementary Data 7). Moreover, interactions of NADH:ubiquinone oxidoreductase subunits with the respiratory syncytial virus (RSV)[35], human immunodeficiency virus 1 (HIV-1)[36], human gammaherpesvirus 8 (HHV-8)[37], human papillomavirus (HPV) serotype 18[38], hepatitis C virus (HCV)[39] and Nipah virus[40] have been reported. An approved drug that inhibits NADH:ubiquinone oxidoreductase is the biguanide metformin[41], the first-line treatment in type II diabetes[41]. Intriguingly, previous observational studies have shown reduced mortality of

type II diabetic patients taking metformin during COVID-19[42]. The aim to improve the effect of metformin led to the development of phenformin[43], which showed an improved cellular uptake compared to metformin[44]. We hence decided to use phenformin for experimental validation of our approach. The solute carrier family 16 member 1 gene (SLC16A1), a monocarboxylic acid-transporter that transports compounds such as pyruvate, lactate, branched-chain amino acids, and ketone bodies across the cell membrane[45], interacts with SARS-CoV-2 and influenza A virus (IAV) proteins. It was additionally identified as a host cell factor of HCV[46] and demonstrated to be a relevant host factor for SARS-CoV-2 infections in two CRISPR-Cas9 knockout screens[33,34]. SR13800 was identified as a potential inhibitor of SLC16A1[47,48]. The succinate-ubiquinone oxidoreductase subunit A (SDHA) is part of complex II of the respiratory chain and catalyzes the reaction of succinate to fumarate in the tricarboxylic acid cycle. Besides its interaction with SARS-CoV-2[33,49], SDHA also interacts with the Epstein-

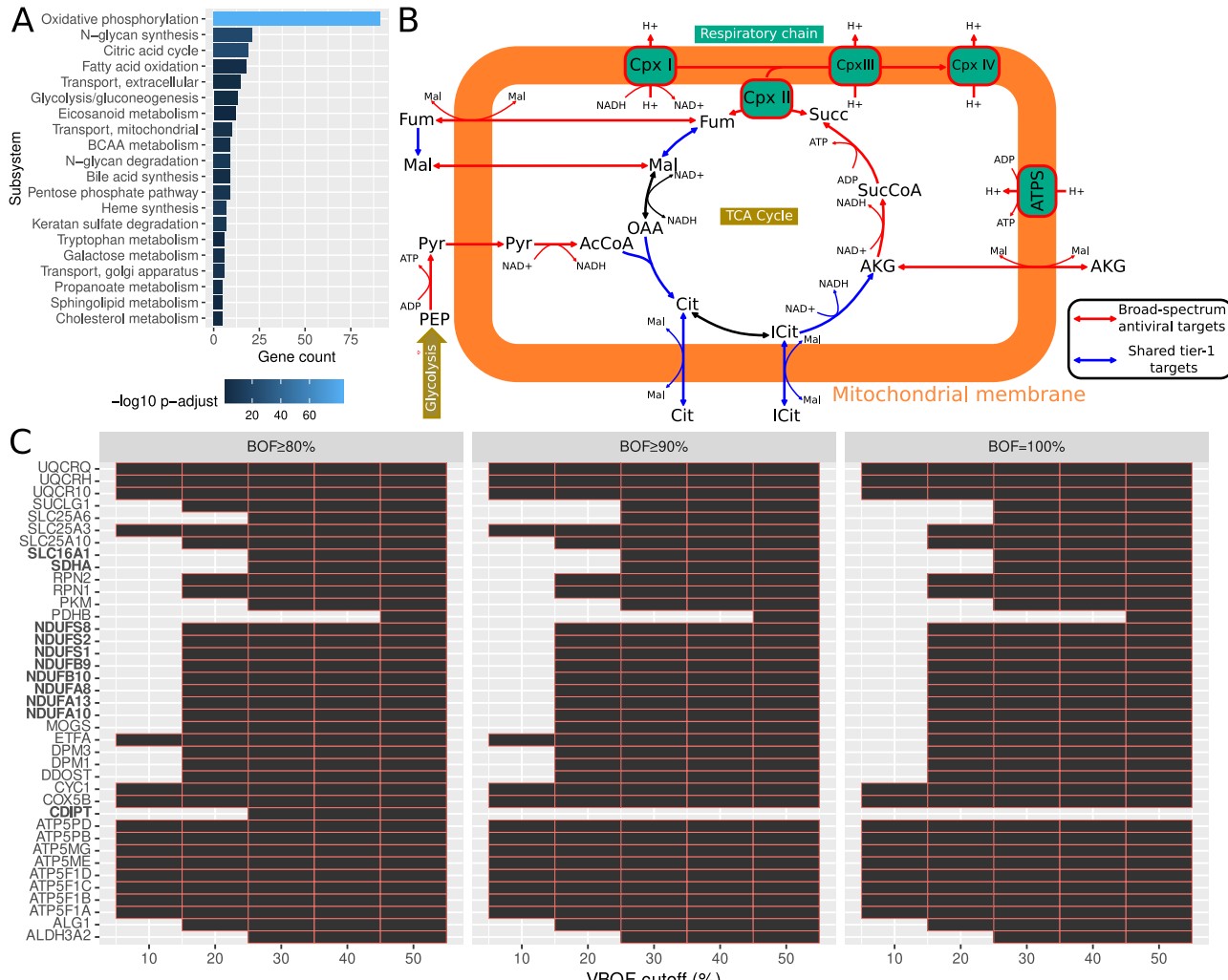

**Fig. 3 | Shared tier-1 and broad-spectrum antiviral targets. A** Enrichment of metabolic subsystems among shared tier-1 targets. The 20 most significantly enriched metabolic subsystems are shown. Abbreviations: BCAA metabolism, branched-chain amino acid metabolism. **B** Shared tier-1 targets and broad-spectrum antiviral targets in central metabolic pathways. Broad-spectrum antiviral targets are also shared tier-1 targets. Please note that transport reactions involving malate are anti-porters. See Supplementary Data 1 for a complete map of human metabolism. Abbreviations: AcCoA acetyl-CoA, AKG alpha-ketoglutarate, ATPS ATP synthase, Cit citrate, Cpx I–IV complex I–IV, Fum fumarate, ICit isocitrate, Mal malate, PEP phosphoenolpyruvate, Pyr pyruvate, Succ succinate, SuccCoA succinate-CoA. **C** Sensitivity analysis of identified broad-spectrum antiviral targets. Cut-offs used for the analysis are indicated in the plots. Black rectangles represent genes that are identified as broad-spectrum antiviral targets in the corresponding setting. Bold gene names correspond to those selected for experimental validation.

Barr virus (EBV)[38], the IAV[50], HCV[51], and HHV-8[37]. Atpenin A5 is a known SDHB inhibitor[52] and was previously assessed for its therapeutic effect on cardiac injury in isolated perfused rat hearts[53]. The CDP-diacylglycerol-inositol 3-phosphatidyltransferase (CDIPT) is involved in the biosynthesis of phosphatidylinositol. It interacts with SARS-CoV-2 as well as IAV and is reported in three CRISPR-Cas9 studies as a relevant host factor during SARS-CoV-2 infection[32–34]. CDIPT also interacts with proteins of numerous HPV serotypes[38,54], EBV[38], and HIV-1[55]. Scyllo-inositol is listed as an inhibitor of CDIPT[56] and was proposed as a potential drug to prevent Alzheimer's disease[57].

## Experimental validation of broad-spectrum antivirals in viral infection assays

The antiviral activity of the identified compounds phenformin, atpenin A5, SR13800, and scyllo-inositol was first tested in the lung-cancer cell line Calu-3 and the colon-cancer cell line CaCo-2 infected with SARS-CoV-2 (Fig. 4A). The infection rate was measured by virus-encoded reporter gene expression (mNeonGreen) 48 h after infection. The cell viability was assessed by Hoechst staining of the nucleus (see "Methods") and hence assessment of total cell numbers after 72 h of treatment, which is a good proxy for cell survival and cell growth upon prolonged compound exposure. Phenformin and atpenin A5 inhibited SARS-CoV-2 infection of Calu-3 cells at different doses and partly of CaCo-2 cells, whereas SR13800 showed toxicity at higher concentrations (Fig. 4B). Scyllo-inositol showed no activity (Fig. 4A and representative primary images in Fig. 4C).

Phenformin and atpenin A5 showed antiviral activity against mNeonGreen expressing SARS-CoV-2, which is based on an early Wuhan isolate (Fig. 4A). Since we identified both compounds in silico as potential broadly acting antivirals, we assessed their $IC_{50}$ in inhibiting SARS-CoV-2 including an Omicron variant, dengue virus (DENV), the respiratory syncytial virus (RSV) and IAV (Fig. 5A) and the $CC_{50}$ based on total cell numbers (Fig. 5B). Phenformin had an $IC_{50}$ of 1.81 μM, and atpenin A5 an $IC_{50}$ of 0.45 μM against the original Wuhan strain in Calu-3 cells without cellular toxicity and both compounds were similarly active against the Omicron variant. It was previously reported that type II diabetic patients taking metformin, similar to phenformin, showed a reduced risk of severe COVID-19 compared to non-users[42]. We also included metformin in our SARS-CoV-2 infection experiments revealing that it inhibited infection only at very high concentrations at an $IC_{50}$ of 459 μM in Calu-3 cells, again without showing any cytotoxicity (Fig. 5 and Supplementary Fig. 1). We

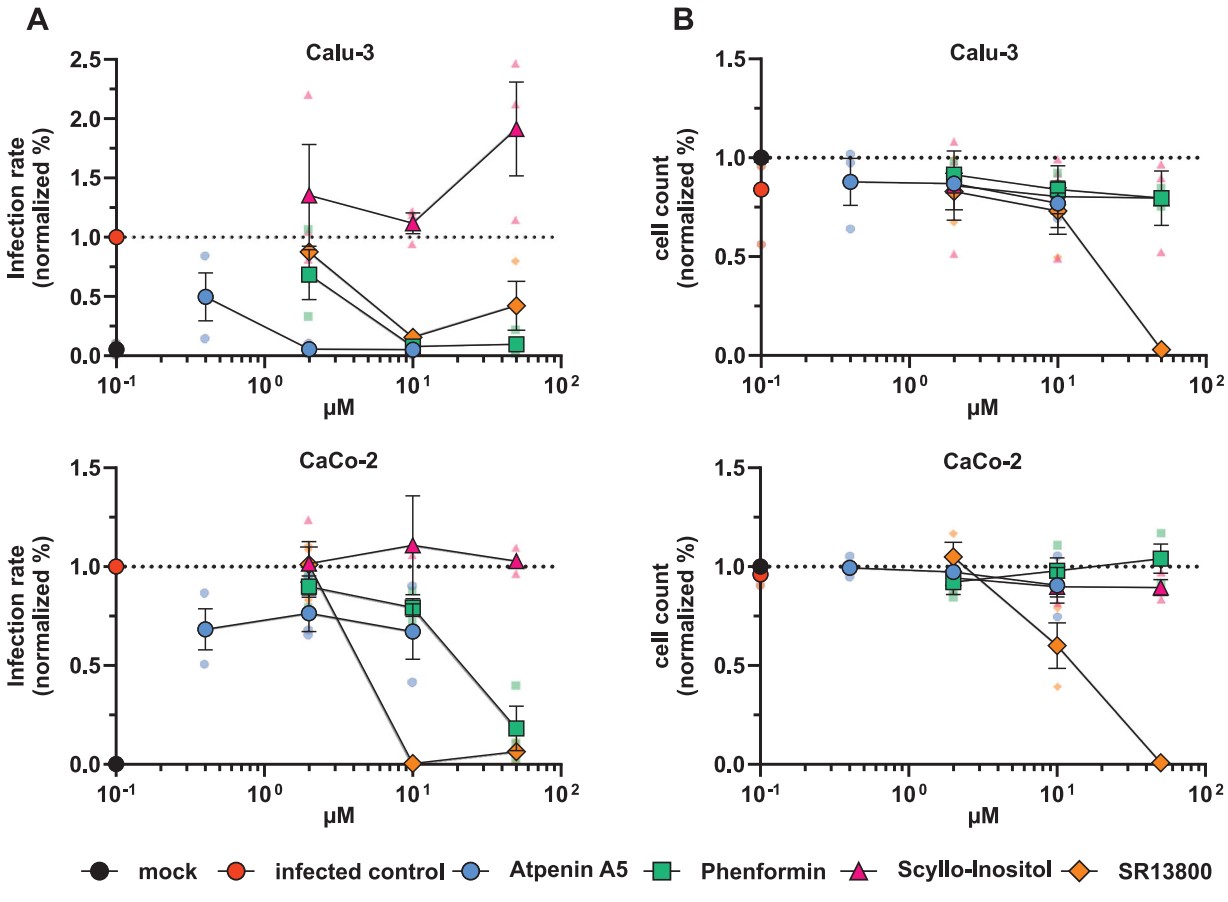

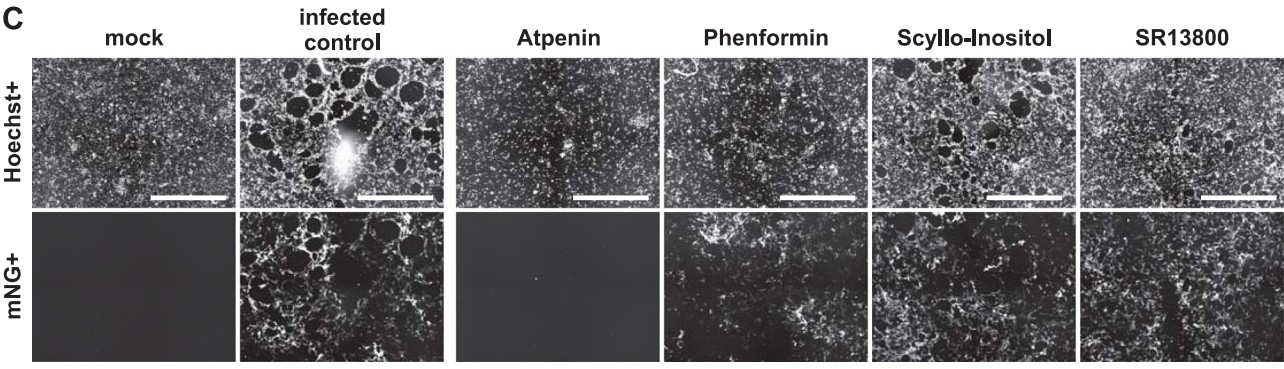

**Fig. 4 | Atpenin A5 and phenformin show antiviral activity against SARS-CoV-2.**
Calu-3 and CaCo-2 cells were pretreated for 24 h with atpenin A5, phenformin, scyllo-inositol, and SR13800 in indicated concentrations and infected with icSARS-CoV-2-mNG (MOI 0.2 for CaCo-2 and MOI 0.5 for Calu-3). Forty-eight hours post-infection, cells were fixed, stained with Hoechst, and measured with a Cytation3 multiplate reader. $n = 3$. Data represent means ± S.E.M. **A** Infection rate (mNG

+/Hoechst+ cells) normalized to control/DMSO-treated infected cells. **B** Overall cell count (Hoechst+ cells) normalized to mock. **C** Representative fluorescence microscopy images were taken at 4-fold magnification to detect cell nuclei count (Hoechst+) and infected cells count (mNeonGreen+). Shown are Calu-3 cells treated with 2 μM of the respective compounds. Scale bar is 1 μm.

therefore decided to do further tests exclusively with phenformin, as metformin was clearly inferior in its antiviral activity.

Phenformin had an $IC_{50}$ of 17.15 μM against DENV in Huh7.5 hepatoma cell lines, while atpenin A5 inhibited DENV infection at an $IC_{50}$ of 0.38 μM (Fig. 5A and Supplementary Fig. 1). Of note, atpenin A5 also inhibited RSV and IAV infection of the lung cell line A549 at $IC_{50}$s of 1.01 μM (IAV) and 0.12 μM (RSV), respectively (Fig. 5A and Supplementary Fig. 1). Phenformin was inactive against RSV and IAV at the concentrations tested. We did not measure any severe cytotoxicity of any of the compounds in the three cell lines tested (Fig. 5B).

**Detailed analysis of atpenin A5 and phenformin cytotoxicity and inhibition of metabolic activity in various cell lines**
As predicted from our modeling approach as tier 1 targets, atpenin A5 at 20 μM and phenformin at 100 μM do not show toxicity in endpoint measurements based on total cell counting when Calu-3, CaCo-2, A549, and Huh7.5 cells were exposed to the compounds for 72 h (Fig. 4 and Fig. 5). However, since cell counting is only a rough proxy to assess cytotoxicity and a higher dose escalation is necessary for $CC_{50}$ determination, we further included careful assessment of growth kinetics by live cell imaging, measurement of mitochondrial metabolic activity by MTT assay as well as

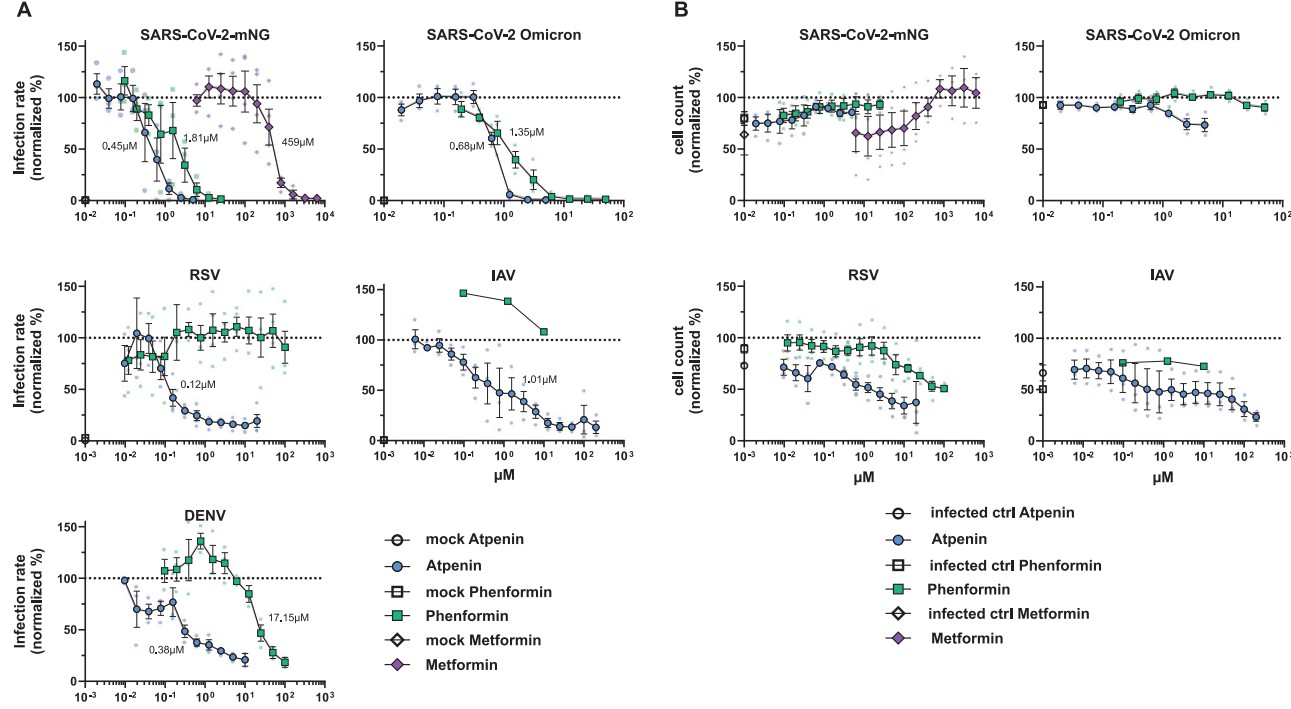

**Fig. 5 | IC$_{50}$ determination against different RNA viruses in various cell lines of atpenin A5 and phenformin.** Atpenin A5 and phenformin were tested for antiviral activity against two SARS-CoV-2 strains, icSARS-CoV-2-mNG (MOI 0,5), and the variant of concern Omicron (MOI 1.1), a respiratory syncytial reporter virus (RSV, MOI 0.6) and an influenza A reporter virus (IAV, MOI 1.4) expressing GFP as well as a luciferase-expressing dengue reporter virus. Cells were pretreated with the compounds for 24 h, respectively for 1 h (DENV, MOI 1). Forty-eight hours post-infection, cells were fixed, stained with Hoechst, and measured with a Cytation3 multiplate reader. DENV-infected cells were lysed to measure luciferase activity. **A** Graphs show the infection rate normalized to untreated infected cells. μM values in the graphs refer to the specific IC$_{50}$ values. **B** Cell count (Hoechst+ cells) normalized to mock. $n = 3$; $n = 1$ (IAV, phenformin). Data represents means ± S.E.M.

cellular ATP levels by CellTiter-Glo (CTG, see "Material and Methods" section for details). The data is summarized in Fig. 6. Careful titration of atpenin A5 indicated CC$_{50}$ values based on growth curve assessment in Calu-3 cells > 160 μM, 115 μM in Huh7.5, and 67 μM in A549 cells (Fig. 6A). In comparison, phenformin showed much less impairment of cell growth with a CC$_{50}$ of 2060 μM in Calu-3 and 1313 μM in Huh7.5 cells (Fig. 6B). As expected, since MTT and CTG assays measure cellular metabolism, the CC$_{50}$ values of both compounds were somehow lower in the various cell types as compared to growth curve assessment. This can be interpreted as a surrogate for the compounds' on-target activity (Fig. 6C). CC$_{50}$ values measured for atpenin A5 based on MTT and CTG closely mimic the IC$_{50}$, with 3.6 μM (MTT) and 1.0 μM (CTG), respectively. The other cell types, Huh7.5 and A549, seem to have more active metabolic pathways showing only modest impairment in MTT and CTG in the presence of atpenin A5. Similarly, phenformin showed a lower on-target activity in both assays than atpenin A5 (Fig. 6C), as indicated by its higher IC$_{50}$ values.

Finally, determination of the IC$_{50}$ (Fig. 5) and CC$_{50}$ (Fig. 6) allowed us to calculate the selectivity indices (SI) of atpenin A5 and phenformin against different viruses in the various cell lines (Table 1). We conclude that atpenin A5 is a potent antiviral that inhibits SARS-CoV-2, DENV, IAV, and RSV with SIs throughout >100 when considering growth kinetics as the most relevant assay to determine impairment of cellular viability. Phenformin potently inhibited SARS-CoV-2 and DENV, while it was not active against IAV and RSV. Altogether, this data shows that phenformin and atpenin A5 are anti-virals active against several non-related viruses and support the suitability of our workflow to identify broad-spectrum antiviral drugs and targets.

## Compound time-of-addition (TOA) analysis suggests differential modes of antiviral activity

In order to get first insights into the potential mode-of-action exerted by the two compounds against the viruses investigated, we performed time-of-addition assays. Compounds were added from 24 h before infection up to 2-h post infection. Hypothesizing that the compounds suppress metabolism and therefore block viral gene expression, adding the inhibitors 2 h after infection, when entry has occurred, should not have a dramatic negative impact on their antiviral activity. Adding atpenin A5 inhibited SARS-CoV-2, RSV, IAV, and DENV as expected (Fig. 7). However, of surprise, atpenin A5 showed a clear TOA effect against SARS-CoV-2 and RSV, even though it still exerted some antiviral activity when added 2 h post infection. This suggests that the compound has a dual mode-of-action against these two viruses, likely involving inhibition of metabolism as well as entry. In contrast, there was clearly no TOA effect of atpenin A5 in suppressing IAV and DENV infection, indicating that these two viruses are inhibited by the compound's action on metabolism. A similar phenotype was observed for phenformins antiviral activity on SARS-CoV-2 versus DENV (Fig. 7B). In conclusion, while TOA is compatible with the known metabolic suppression of phenformin and atpenin A5, there might be additional antiviral activity exerted by both compounds.

## Antiviral activity of phenformin in vivo

Given that phenformin is an established drug in humans with known pharmacokinetics and safety profiles, that is hence readily available, we further evaluated the in vivo antiviral potential of phenformin against SARS-CoV-2 in a Golden Syrian hamster model of infection (Fig. 8A). Hamsters have been identified as an effective in vivo model for SARS-CoV-2 due to their physiological and immunological parallels to humans, enabling a comprehensive analysis of viral replication, transmission, and disease progression[58]. Moreover, developing severe, human-analogous symptoms in hamsters following infection with SARS-CoV-2 closely mirrors the clinical manifestations of humans, making them an invaluable model for studying potential therapeutic interventions[59]. Two groups of seven female 9-11 week-old animals were intranasally infected on day 0 (D0) with 10$^{5,5}$

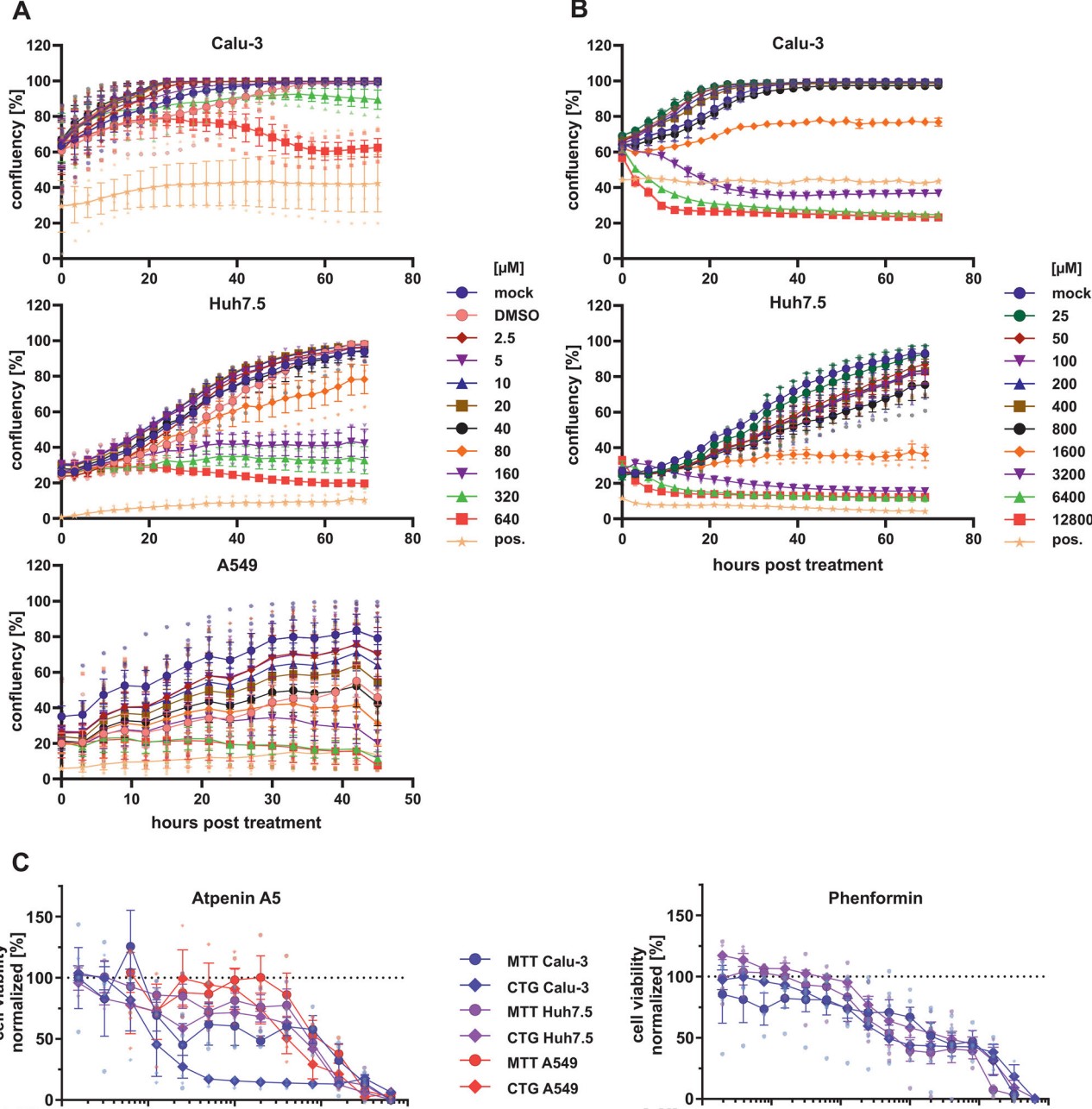

**Fig. 6 | Effects of atpenin A5 and phenformin on the growth and viability of various cell lines.** Calu-3, Huh7.5, and A549 cells were treated with the indicated concentrations of **A** atpenin A5 and **B** phenformin for 48 h (A549) or 72 h (Calu-3, Huh7.5). DMSO (1.3%) is a solvent control. The positive control is 50% DMSO to inhibit cell growth. Plates were imaged in the IncuCyte® at 37 °C, 5% CO$_2$, every 3 h to assess cell confluency. For analysis, IncuCyte® S3 Software was used. Number of replicates: Calu-3, $n = 2 / 3$; Huh7.5, $n = 3$; A549, $n = 5$. Data represent means ± interplate S.E.M. **C** Cellular metabolic activity was measured using CellTiter-Glo® or MTT assay as described in the "Material and Methods" section. Graphs show normalized MTT signal to non-treated cells or normalized luminescence signal to non-treated cells. $n = 3$; Data represent means ± S.E.M.

TCID50 of a SARS-CoV-2 Delta (B.1.617.2) strain. The "Treated" group received a 5-day oral treatment course of 100 mg/kg phenformin (once daily for 5 days, D-2 to D2), whereas the "Vehicle" group received sterile water. A third group of two "Uninfected Treated" animals received the same phenformin regimen as the "Treated" group but was not infected. Animals were monitored for clinical signs, and oropharyngeal swabs were performed daily to assess viral load in the upper respiratory tract. Lung viral titers were evaluated in a subset of animals on D3 and D6. Of note, two animals of the "Treated" group were lost due to technical problems during gavage, hence reducing to 5 the effective n of this group and precluding statistical analysis beyond D3.

Both "Treated" and "Uninfected Treated" groups showed higher yet mild mean maximum weight losses on D2 compared to the "Vehicle" group (9.5%, 8.7%, and 3.4%, respectively, compared to the baseline established on D-3 before treatment start). Besides, no other behavioral or clinical differences were observed among the three tested groups (Fig. 8B). Infection with the SARS-CoV-2 Delta (B.1.617.2) strain resulted in rapidly detectable viral genome copies in oropharyngeal swabs of animals from the "Vehicle" group (Fig. 8C). Viral titers plateaued during the first 3 days post-infection ($6.9 \pm 9.9 \times 10^2$, $6.0 \pm 7.5 \times 10^2$, $6.0 \pm 6.6 \times 10^2$ nsp14 copies/ng of RNA on D1, D2, and D3, respectively) and then progressively decayed close to the limit of detection by D6. Phenformin treatment significantly reduced viral

replication, as shown by lower mean oropharyngeal viral titers in the "Treated" group $(1.1 \pm 0.9 \times 10^2,\ 0.8 \pm 1.3 \times 10^2,\ 1.5 \pm 1.2 \times 10^1$ nsp14 copies/ng of RNA on D1, D2, and D3, respectively) compared to those of the "Vehicle" group. Moreover, no viral genomes could be detected in samples from the "Treated" group on D5 and onwards. No significant differences were observed in mean lung viral titers between the "Vehicle" $(3.9 \pm 1.8 \times 10^7$ and $1.6 \pm 0.4 \times 10^4$ nsp14 copies/g of lung tissue on D3 and D6, respectively) and "Treated" $(3.3 \pm 4.8 \times 10^7$ and $1.4 \pm 0.4 \times 10^4$ nsp14 copies/g of lung tissue on D3 and D6, respectively) groups (Fig. 8D). As expected, no viral genome was detected in the "Uninfected Treated" group.

Altogether, our in vivo results validate the capacity of phenformin treatment to reduce viral load and shorten time to negativity in the upper respiratory tract of Golden Syrian hamsters. The fact that we did not observe an antiviral effect in the lower respiratory tract underscores the need of further treatment optimization before its potential evaluation in humans.

## Discussion

In this work, we have used constraint-based metabolic modeling to investigate the metabolic state of virally infected cells and identify potential targets for broad-spectrum antiviral treatment. We found that viral infection of a tissue led to pronounced induction of predicted viral replication capacity even in non-infected cells. This indicates that SARS-CoV-2 might induce transcriptional programs in neighboring cells that prime non-infected cells for viral replication along with a pronounced induction of many metabolic pathways, which we previously reported for several types of immune cells in COVID-19 patients[60]. Such pleiotropic effects in the microenvironment of virally infected cells have previously been reported for other viruses such as Epstein-Barr Virus and Kaposi's sarcoma herpesvirus[61]. In this context, those effects are mediated via extracellular vesicles secreted by virally infected cells that contain viral effectors that modulate the metabolic activity of non-infected cells in the microenvironment. Our observation of a widespread induction of pathways required for viral replication in uninfected cells, in combination with the previous detection of extracellular vesicles containing viral proteins in SARS-CoV-2 infected individuals[62,63] indicate that SARS-CoV-2 infected cells might modulate the permissibility of their microenvironment for viral replication in a similar manner. Overall, these results suggest that SARS-CoV-2 strongly relies on a profound induction of metabolic pathways for its replication and support the notion that the cellular metabolism of virally infected cells is an attractive target for inhibiting viral replication.

Thus, we used our modeling approach on an expanded set of viruses additionally including DENV, a member of the Flaviviridae and a fast-spreading insect-borne disease, as well as IAV, belonging to the Paramyxoviridae to identify potential targets for broad-spectrum antiviral therapy. Focusing on targets that impeded viral replication with only a small impact on cellular replication (tier-1 targets), we identified 254 potential enzyme targets, of which 158 have already been reported in association with viral proteins. Those enzyme targets were strongly enriched in pathways known to be highly relevant for viral replication such as the electron transfer chain, the tricarboxylic acid cycle and glycolysis[11,28,31]. Further including information about physical interactions of the targets with viral proteomes, we identified 39 proteins forming parts of 22 protein complexes that could serve as druggable targets for broad-spectrum viral inhibition. Among those candidates, we selected four enzyme complexes comprising eleven identified targets for further experimental validation based on the availability of suitable inhibitors. Thus, we could confirm that targeting the NADH:ubiquinone oxidoreductase complex by phenformin and the succinate dehydrogenase by atpenin A5 blocked viral replication in cell culture experiments with minimal cellular toxicity. Remarkably, while phenformin inhibited the replication of DENV and SARS-CoV-2, atpenin A5 showed excellent antiviral activity against DENV, SARS-Cov-2, RSV, and IAV. Furthermore, TOA-assays indicate differential modes-of-action of the compounds used. While DENV and IAV seem mainly inhibited by the compound's effect on metabolism, there is a clear TOA effect of SARS-CoV-2 inhibition by phenformin and atpenin A5 and also RSV is potently

**Table 1 | Selectivity indices of atpenin A5 and phenformin against various pandemic RNA viruses in different cell lines**

| Atpenin A5 | IC$_{50}$ | CC$_{50}$ | selectivity index: CC$_{50}$/IC$_{50}$ |
|---|---|---|---|
| ic-SARS-CoV-2-mNG (in Calu-3 cells) | 0.45 µM CI 95% [0.249–1.361] | MTT: 3.6 µM CTG: 1.0 µM IncuCyte: >160 µM | MTT: 8 CTG: 2.2 IncuCyte: >355 |
| SARS-CoV-2 Omicron (in Calu-3 cells) | 0.68 µM CI 95% [0.620–0.788] | MTT: 3.6 µM CTG: 1.0 µM IncuCyte: > 160 µM | MTT: 5.3 CTG: 1.5 IncuCyte: >235 |
| Dengue Virus (in Huh7.5 cells) | 0.38 µM CI 95% [0.225–0.683] | MTT: 83.3 µM CTG: 85.4 µM IncuCyte: 115 µM | MTT: 219.2 CTG: 258.8 IncuCyte: 319.4 |
| Influenza A Virus (in A549 cells) | 1.01 µM CI 95% [0.561–1.82] | *MTT:* 114.7 µM *CTG:* 49.5 µM *Incucyte:* 67 µM | *MTT:* 113.5 *CTG:* 49 *Incucyte:* 66.3 |
| RSV (in A549 cells) | 0.12 µM CI 95% [0.076–0.190] | MTT: 114.7 µM CTG: 49.5 µM IncuCyte: 67 µM | MTT: 955.8 CTG: 412.5 Incucyte: 558.3 |
| **Phenformin** | **IC$_{50}$** | **CC$_{50}$** | **Selectivity index: CC$_{50}$/IC$_{50}$** |
| ic-SARS-CoV-2-mNG (in Calu-3 cells) | 1.81 µM CI 95% [1.034–3.088] | MTT: 456 µM CTG: 26.3 µM IncuCyte: 2060 µM | MTT: 251.9 CTG: 14.5 IncuCyte: 1138 |
| SARS-CoV-2 Omicron (in Calu-3 cells) | 1.35 µM CI 95% [0.798–1.87] | MTT: 456 µM CTG: 26.3 µM IncuCyte: 2060 µM | MTT: 337.8 CTG: 19.5 IncuCyte: 1525.9 |
| Dengue Virus (in Huh7.5 cells) | 17.15 µM CI 95% [10.21–59.66] | MTT: 64.2 µM CTG: 80.2 µM IncuCyte: 1313 µM | MTT: 3.7 CTG: 4.7 IncuCyte: 76.6 |
| Influenza A Virus (in A549 cells) | *not active* | | |
| RSV (in A459 cells) | *not active* | | |

suppressed by atpenin A5, especially when it is given before infection. Such a TOA-effect is supportive for a compound mode-of-action related to early steps in the viral cycle, i.e., entry. Of note, SARS-CoV-2 and RSV were also sensitive towards inhibition when the compounds were given after infection, indicating that in addition to the effects on entry, metabolism is also relevant for the observed antiviral activity. Altogether, phenformin and atpenin A5 that target pathways based on our predictions, inhibit four viruses belonging to completely distinct viral taxons. This supports the suitability of our pipeline and approach to identify broad-spectrum antivirals. Importantly, our target prediction approach did not incorporate data from RSV-infected cells, thus further supporting the notion that the predicted targets also work beyond the three viruses initially included in our analysis.

Phenformin was well tolerated up to high concentrations, which agrees with its previous use as an antidiabetic drug before it was withdrawn from the market in the 1970s due to more frequent fatal cases of lactic acidosis compared to the alternative antidiabetic metformin. The frequency of lactic acidosis was 40–64 per 100,000 patients per year and probably related to renal insufficiency similar to metformin therapy[64]. However, provided that treatment in the context of acute viral infections is short-termed and therefore fundamentally different from the long-term treatment in diabetes, phenformin could represent a viable treatment option. Indeed, biguanides including phenformin showed clinical benefit in diabetic influenza patients

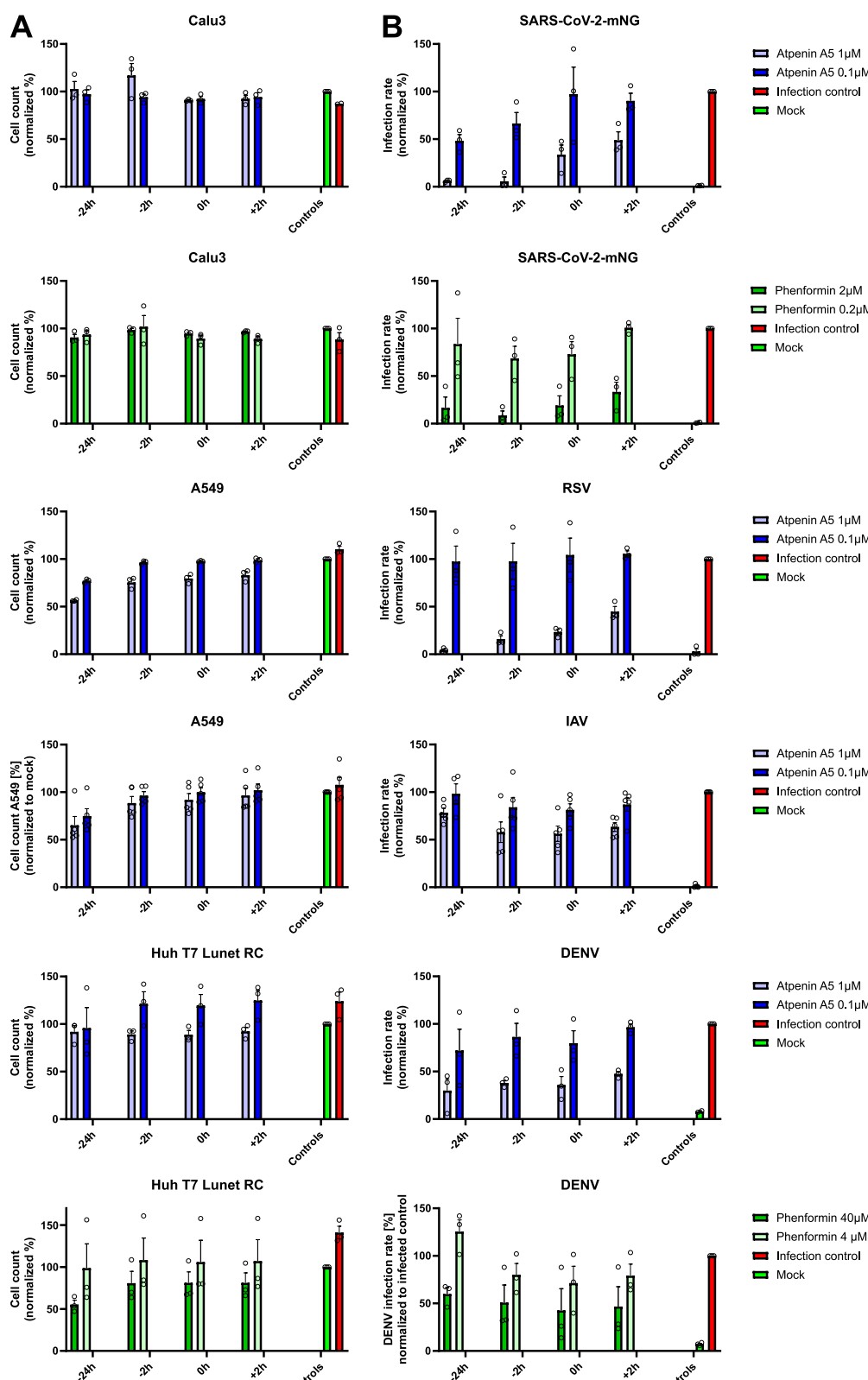

**Fig. 7 | Atpenin A5 and phenformin time of addition assay against different RNA viruses.** Cells (Calu-3, A549, or Huh T7 Lunet RC) were treated with different concentrations of atpenin A5 or phenformin at several timepoints (24 h before infection, 2 h before infection, at the same time or 2 h after infection). They were infected with either icSARS-CoV-2-mNG (MOI 0.3), respiratory syncytial reporter virus expressing GFP (RSV, MOI 0.3), influenza A reporter virus (IAV, MOI 0.3) expressing GFP, or Dengue virus (WT2, MOI 1). Forty-eight hours post-infection, cells were fixed with 2% PFA, stained with Hoechst, and infection rates were measured with a Cytation3 multiplate reader. **A** Cell count (Hoechst+ cells) normalized to mock. **B** Graphs show the infection rate normalized to untreated infected cells. $n = 3$; Data represents means ± S.E.M.

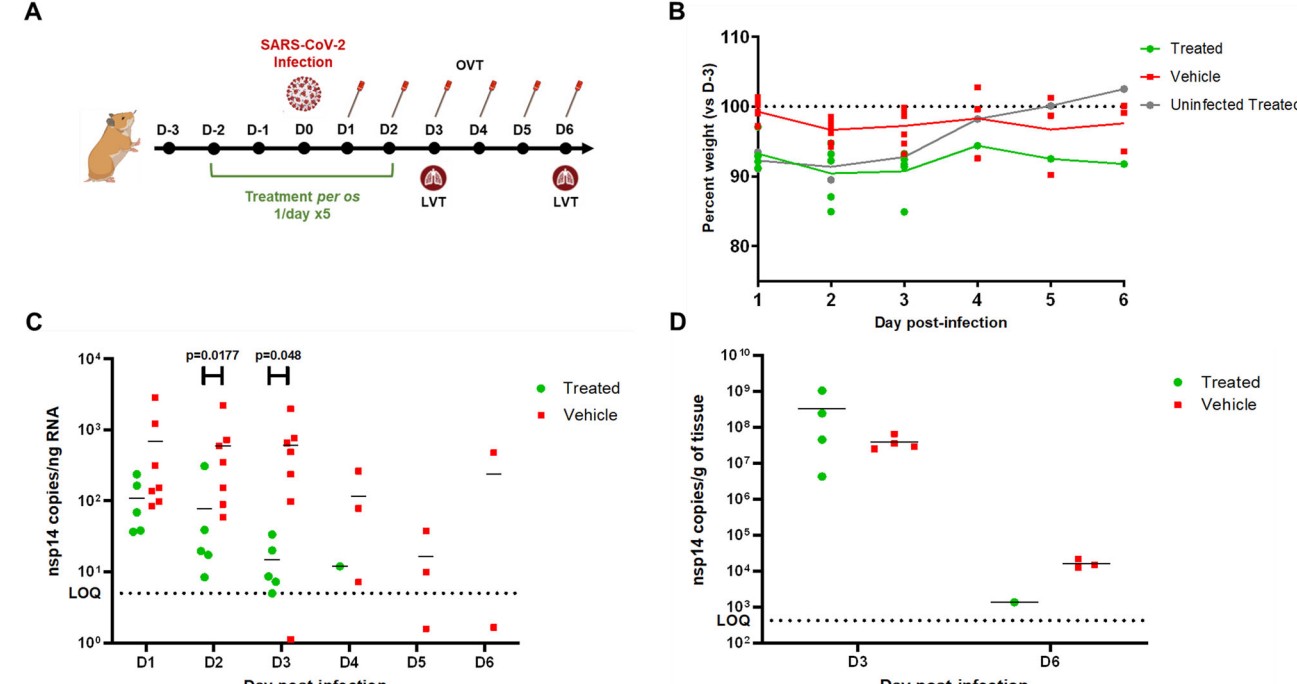

**Fig. 8 | Antiviral activity of phenformin in vivo. A** Golden Syrian hamsters were infected intranasally on day 0 (D0) with $10^{5.5}$ TCID50 of SARS-CoV-2 Delta (B.1.617.2) virus. "Treated" (n = 5) and "Uninfected Treated" animals received 100 mg/kg phenformin by gavage once daily for 5 days, starting on D-2. "Vehicle" animals (n = 7) received PBS. **B** Weight loss was monitored on D-3 (baseline before treatment) and daily from D0 to D6. Data for individual animals (dots and squares) and means (lines) are shown. **C** Oropharyngeal swabs were collected daily from D1 to D6 and viral titers were assessed by RT-qPCR. **D** On D3 and D6 a subset of animals were euthanized and lungs were harvested to assess viral loads by RT-qPCR. Data are shown as means ± SD. Two-tailed p values were determined by Mann-Whitney U test, using GraphPad Prism software v10.3.1. OVT oropharyngeal viral titers, LVT lung viral titers, LOQ limit of quantification.

and hence the compound was suggested as a COVID treatment option[65]. Another study identified phenformin as a potential SARS-CoV-2 PL^pro inhibitor with potential antiviral activity using molecular dynamics simulations[66]. However, none of the aforementioned studies provided experimental evidence for antiviral activity of phenformin in cell-based or in vivo infection models. Reported plasma levels of phenformin after 50-100 mg of phenformin uptake in patients are ~1.7 µM[67], while tissue concentrations have not been assessed but are expected to be much higher as reported in animal models[68]. For instance, in rats portal vein concentrations reached 2.5 µM and liver concentrations 147 µM after oral gavage of 50 mg/kg phenformin[69]. Thus, tissue concentrations are expected to be much higher than the $IC_{50}$ of 1.35–1.8 µM that we measured against SARS-CoV-2. Atpenin A5 is, up to now, an experimental compound that was not tested in vivo, possibly due to its expectable multiple adverse effects in various organs. Atpenin A5 is known as a potent complex II inhibitor[70], thus affecting the redox chain at the mitochondrial membrane and overall energy metabolism. These two features are also reflected by our MTT and CTG assays (Fig. 6 and Table 1). Nevertheless, complex II inhibition is also discussed as a druggable metabolic pathway for various cancer types[71]. Novel atpenin A5-derived leads were developed with a much higher on-target specificity[72]. These recent developments, combined with our observation that complex II inhibition via the succinate dehydrogenase is a druggable and highly potent target for broad-spectrum antiviral inhibition, raise the possibility for the rapid establishment of novel broadly acting antiviral drugs.

This notion and the reliability of our whole workflow are supported by our in vivo validation of the antiviral activity of phenformin in the Golden Syrian hamster model. Although further dose-optimization studies are warranted to maximize the antiviral effect in the lower respiratory tract, phenformin showed antiviral activity in the upper respiratory tract, the main anatomic site for viral entry and spread. Hence, early application of phenformin may potentially reduce the time to symptom resolution,

prevent or delay spread to the lower respiratory tract, and also shorten and limit the transmission window. As already discussed, given the well-elaborated PK/ADME/tox profiles of phenformin in humans, we suggest holding phenformin as a candidate drug for preparedness in case of future outbreaks caused by coronaviruses and potentially other pandemic viruses. Furthermore, recent evidence coming from larger trials analyzing the efficacy of the phenformin-related metformin to ameliorate symptoms of long-covid and also giving benefit to patients during acute infection[73,74], supports our findings. Importantly, in our comparative analyses, phenformin strongly outcompeted metformin in terms of its antiviral activity which was ~300 fold higher against SARS-CoV-2 ($IC_{50}$ of ~1.5 µM for phenformin vs ~450 µM for metformin). This difference in efficiency is in line with the improved cellular uptake of phenformin over metformin and the much lower typical therapeutic dosage of phenformin compared to metformin in humans[68].

Nevertheless, there are limitations to our study and open questions remain. On the computational side, we applied our workflow only to single-cell RNA-Seq datasets which come with the limitation that typically only a subset of expressed genes is detected[75]. However, bulk sequencing RNA-Seq data suffers from the disadvantage that it also includes transcripts of genes from cell types that might not be permissible for viral infection. For instance, SARS-CoV-2 is typically only able to infect a small subset of cells in the lung[76]. Moreover, due to the reconstruction of context-specific models for hundreds of thousands of cells being a limiting step in our analysis, we based our approach on Recon2.2[19] which is much smaller in size than more current reconstructions of human metabolism, like Recon3D[77]. Also, the mapping approach for transcriptomic data is based on the inclusion of highly transcribed genes and therefore is biased towards more abundant proteins, hence potentially missing lowly abundant proteins relevant for viral replication. Our subsetting approach might further focus predicted targets to those with known interactions with viral proteins since experimental approaches for detecting such interactions are likely biased to more

abundant proteins. Additionally, by concentrating on targets that can inhibit replication across several viruses, we might have missed potent targets for viral inhibition in individual viruses. The activity of phenformin needs to be characterized in further detail in vitro as well as in vivo prior to upcoming clinical trials. While we observed some weight loss in vivo, this is likely related to an appetite-reducing effect as has been observed for metformin[78], the weight trend reassuringly did not change post-infection. Furthermore, phenformin was not active in vitro against RSV and IAV. For both infection models we employed A549 cells and these lung adenocarcinoma cells might exert cell-line specific resistance to this compound. So it will be essential to further exploit other cell types and foremost primary cell models, for example air-liquid-interface cultures to verify this phenotype. Atpenin A5 is of limited in vivo compatibility, and following its proof-of-concept as broad antiviral agents, more work is necessary to render it biocompatible or find other inhibitors of the same target. Moreover, we have only explored a subset of our predicted targets, and other targets from our list might provide an even higher potency for inhibiting viral replication in a broad set of viruses. Hence, ample follow-up studies to the work presented herein are necessary to further explore the capability of our modeling workflow to predict antiviral drugs.

Altogether, this study proves that targeting metabolism is a valuable strategy in the context of antiviral therapies[79], with certain advantages. Metabolism-based targets exert a very low variability and are predictably essential for viral replication, resulting in a high resistance barrier and a predictable broad antiviral activity. In this context, our workflow for identifying antiviral targets, integrating cellular metabolism and data from virally infected cells, as well as the inhibitors we have identified, represents an invaluable resource for pandemic preparedness against future emerging pathogens.

## Methods

### Computational analysis—single-cell sequencing datasets
Table 2 summarizes the single-cell sequencing datasets used.

### Computational analysis—viral replication models based on Recon 2.2
The viral replication reaction for SARS-CoV-2 was taken from Renz et al.[17]. The viral replication reaction for dengue was taken from Aller et al.[16]. No viral replication reaction for influenza A H1N1 was available and, thus, needed to be constructed. The nucleotide and protein sequences with the RefSeq number GCF_001343785.1 were downloaded from NCBI's Reference Sequence database[80]. The genome copy number was assumed to be 1. Influenza A H1N1 has four structural proteins: hemagglutinin (HA), neuraminidase (NA), matrix protein 1 (M1), and matrix protein 2 (M2). The nonstructural proteins include the polymerases PB1, PB2, and PA, the nucleocapsid protein, the nonstructural protein 1, the nuclear export protein, and the PA-X protein. The detailed list of copy numbers for the structural and nonstructural proteins is given in Supplementary Data 9. The stoichiometric coefficients for the nucleotides, amino acids, and energy requirements were calculated based on the copy numbers and according to the steps suggested by Aller et al.[16]. These coefficients are provided in Supplementary Data 10. We did not consider lipids for the viral biomass

reaction since for most viruses there is only scarce information about the composition of the viral lipidome[81] and this has just recently been determined for SARS-CoV-2[82]. So far, detailed knowledge about virus envelopes remains limited in most cases, making it challenging to include, for instance, lipid requirements in the virus biomass objective functions. Later generations of models should consider this information when it becomes available. A recent study yielded additional model predicted targets after including lipid requirements[83]. Please note that due to the modeling procedure inclusion of lipid metabolites will only increase the number of targets. The genome-scale metabolic model of humans, Recon 2.2[19], was expanded with the viral replication reaction for each of the three viruses. The model was conditioned with a medium corresponding to the concentration of metabolites present in human blood (Supplementary Data 11)[60]. Please note that this diet approximates maximal uptake rates for metabolites based on their concentration, as typically done when considering complex media[84]. Subsequently, we used fastcore's fastcc algorithm and the simulated blood serum diet to generate consistent models for each virus. These models are available from the BioModels database (see "Data Availability").

### Computational analysis—cell identity annotation
Cell-type annotation was not available for all datasets analyzed in this study. For deposited data with missing metadata on cell identity, we performed cell type annotation de novo (see Supplementary Fig. 2A–D). For the datasets BALF1 and dengue, we employed the R-package CHETAH (version 1.8.0)[85]. CHETAH makes use of a reference single-cell RNA dataset, which is used to build a classification tree via hierarchical clustering. Input cells are placed and identified as reference cell types or intermediate types in the classification tree. The *BALF1 dataset* was annotated using a single-cell lung atlas of idiopathic pulmonary fibrosis, chronic obstructive pulmonary disease, and healthy smoker and non-smoker's lung cells (ref.[86], GSE136831). Only cells annotated as epithelial in the original work were considered.

The *dengue dataset* was annotated using an in-house available PBMC single-cell dataset[87]. References were prepared according to the CHETAH R-package manual. Briefly, annotation references were created from single-cell expression datasets with known cell type identifications. Single cells of only healthy donors were kept in the reference. Rare cell types with less than 15 cells representing that cell type were dropped from the reference. In the interest of computing times, the remaining cell types were downsampled to a maximum of 300 cells, picked at random. The remaining single cells were normalized sample-wise to an equal sequencing depth of one million, and +1 was added to each gene's count to allow for $\log_2$ transformation. For an improved classification, ribosomal and housekeeping genes were dropped (Supplementary Data 12). After creating Seurat objects and performing quality control as described in "scRNA core reaction pre-processing," cell type annotation was done on the Seurat objects. For this, the classification function CHETAHclassifier was run with the query dataset and the matching reference. The quality of the automated classification was visually controlled with dimension reduction plots.

The *influenza H1N1 dataset* was manually annotated since CHETAH provided unsatisfactory results. We followed the steps outlined in the accompanying publication's methods[88].

## Table 2 | Single-cell sequencing datasets

| Dataset | Reference | Accession Number | Celltype annotation source | # CS models |
|---|---|---|---|---|
| BALF1 | 26 | GSE145926 | Annotation via CHETAH version 1.8.0 with single cell lung atlas (GSE136831[86]) | 1678 |
| BALF2 | 22 | EGAS00001004481 | Original authors (https://doi.org/10.6084/m9.figshare.12436517) | 148,420 |
| CALU-3 | 27 | GSE148729 | Single cell experiment | 32,003 |
| scH1299 | 27 | GSE148729 | Single cell experiment | 20,604 |
| Influenza H1N1 | 28 | GSE191176 | Manual annotation following methods of original publication | 3066 |
| Dengue | 30 | GSE116672 | Annotation via CHETAH version 1.8.0 with PBMC single cell reference[55] | 64,638 |

The last column indicates the number of context-specific models reconstructed from each dataset.

## Computational analysis—scRNA data processing and reconstruction of context-specific metabolic models

Prior to the reconstruction of single-cell metabolic models, scRNA datasets (Methods: scRNA Datasets) were downloaded from NCBI's Gene Expression Omnibus[89] for pre-processing with Seurat[90] and StanDep[20]. Seurat objects of the respective scRNA datasets were created, and cells were removed if data was of insufficient quality (number of detected genes > 200 and <6000; mitochondrial RNA with mapped reads < 10). For datasets with missing cell type information, cell identity annotation was applied to the filtered Seurat objects (see "Cell identity annotation"). Gene level counts were translated into transcripts per million (TPM) values through normalization according to human ENSEMBL gene lengths. Please note that while we used TPMs, the protocols used for scRNA-Seq often involve sequencing from the start or end of a transcript and thus approaches for normalization not involving gene length might be more suitable for this type of data. ENSEMBL[91] gene names were mapped to Recon 2.2 identifiers[20,92]. StanDep was employed with the pre-processed expression data to identify core reactions active in the individual cells. To this end, enzyme expression was $\log_{10}$ transformed into a matrix with rows representing enzymes and columns as bins to identify minimum and maximum enzyme expression. A complete linkage metric with hierarchical clustering and Euclidean distance was used to cluster (number of clusters = 40) genes with respect to their expression. Core reaction matrices were assembled, defined, and used as an input to fastcore[21] to reconstruct single-cell, context-specific metabolic models. The fastcore algorithm was employed to create cell-type-specific models based on the consistent Recon 2.2 model, including the viral replication reaction and the list of core reactions from *StanDep* expanded by the biomass objective function and viral replication reaction. The human metabolic models with all constraints for the blood medium and the viral replication reactions have been uploaded to the BioModels Database[93] in SBML format[94] with the extensions for hierarchical model composition[95] and flux balance constraints version 2[96]. Each model entry in BioModels Database contains a base model derived from Recon 2 and tissue-specific models, all wrapped together in an Open Modeling EXchange format[97] file with annotation[98]. For links to the respective datasets and models see the "Data Availability" statement.

Reactions identified as being active in single-cell metabolic models were counted and summarized into 82 metabolic pathways based on the subsystem annotation of Recon 2.2 to analyze metabolic pathway activity in the BALF2 data. The small number of single-cell models with detected viral RNA were not considered to avoid confounding (87 of 148.420 cells). The resulting counts of active reactions per pathway were checked for too many single cells with zero counts, e.g., metabolic pathways were discarded if they showed zero counts in more than half of the single-cell models across all three patient groups (control, moderate, and severe COVID). The remaining 57 metabolic pathways were statistically evaluated individually for their relation to patient groups while controlling for cell type ("Basal," "Ciliated," "Ciliated-dif," "FOXN4," "Ionocyte," "IRC," "outliers_epithelial," "Secretory," "Secretory-diff," "Squamous"). Active reaction counts were modeled as the dependent variable in negative binomial regressions (R-package MASS 7.3-57 function glm.nb[99]). Metabolic pathways with inflated zero counts were modeled with negative binomial regression with an additional zero-inflation model. The zero-inflation model took the same independent variables as the primary model, namely patient group and cell type (R-package pscl 1.5.5, function zeroinfl with dist = "negbin"[100]). The logarithm to the base two was calculated for fold changes between the control group as a baseline and any of the two COVID patient groups.

## Computational analysis—prediction of viral replication capacity and antiviral targets

In order to predict viral replication capacity for a cell built from scRNA-Seq data, we used flux balance analysis[17] with the applied blood serum diet and maximized the flux through the VBOF. We performed single gene deletions to predict antiviral targets and assessed their effects on host biomass

production and viral replication capacity. Supplementary Data 2 lists the predicted viral replication capacities and predicted tier-1 and tier-2 targets for each cell. Those gene knockouts that were found to decrease viral replication capacity by at least 50% of the initial value while maintaining the host's biomass minimally at 80% of its initial value were reported as tier-1 targets. Tier-2 targets reduced viral replication capacity by at least 50% but decreased cellular biomass production by more than 20%. While the biomass reaction is typically used to predict maximal growth rates of cells, it consumes all the molecules whose production is continuously required to maintain cellular function and hence can be used as a proxy for metabolic requirements of normal cellular function. Examples of such cellular functions include, for instance, the production of amino acids for protein turnover, nucleotides for DNA repair, and energy equivalents for cellular function. To identify tier-1 and tier-2 targets, we considered only cells with a predicted viral replication rate above 0.01 mmol/g$_{DW}$ (unit of flux measurement in constraint-based models) and detected tier-1 targets. Moreover, we constricted the data to include only virally infected cells where possible (i.e., a sufficient number of virally infected cells). Thus, we only considered infected cells as indicated in the provided metadata for the cell culture data from SARS-CoV-2 infected cells (CALU3, scH1299) and the BALF1 data set. For the BALF2 data set, only a few infected cells were detected. Since we observed priming of non-infected cells for viral replication upon viral infection of the host, we considered all SARS-CoV-2 permissive cell types of SARS-CoV-2 infected individuals for this dataset (i.e., those listed in Fig. 1A). For dengue and influenza A, we considered only cells with >0.1% of reads mapping to the respective viral genomes as indicated in the metadata provided along with the sequencing data.

Please note that applying the used cutoffs (<50% maximal viral replication rate, >80% maximal biomass production rate) on the human metabolic model with our predefined diet without the mapping of transcriptomic data did not yield any tier-1 targets. However, two tier-2 targets could be identified: asparagine synthetase (ASNS) and cytidine/uridine monophosphate kinase 1 (CMPK1). ASNS knockout reduces maximal biomass production rate to 57%, while CMPK1 knockout is lethal. This result highlights the relevance of combining the generic model of viral replication with gene expression data.

For the comparison of known physical interaction partners of the viral proteome with tier-1 and tier-2 targets (Fig. 3), we subsetted tier-1 and tier-2 targets to those occurring in at least 5% of the cells for each data set for which tier-1 targets could be identified (high-confidence tier-1 and tier-2 targets, Supplementary Data 5). To identify broad-spectrum antiviral targets, we required that the corresponding enzyme was identified as a tier-1 target in at least one cell in each dataset, yielding a total of 254 shared tier-1 targets across all datasets/viruses. Additionally, we included information on known virus protein interactions[49,101,102]. For SARS-CoV-2, we used host proteins that had at least one reported interaction with a SARS-CoV-2 protein at a SAINTexpress Bayesian false-discovery rate (BFDR) ≤ 0.05[103] and an average spectral count ≥2. The list of considered interaction partners for each virus is provided in Supplementary Data 4. To analyze the enrichment of predicted shared tier-1 targets across all datasets among cellular metabolic pathways, we used the human metabolic model Recon2.2 to identify genes associated with each metabolic subsystem annotated in the model. Thus, we mapped each subsystem to the reactions that it contained and identified the associated gene lists via the gene-protein-reaction associations. Subsequently, we used Fisher's exact test to determine the significance of the enrichment of shared tier-1 targets among the genes associated with each subsystem.

We first determined the total weight of compounds required for a set flux of 1 mmol/g$_{DW}$/h in the biomass reaction and the viral replication reaction to determine the energetic requirement of normal growth versus viral replication. To this end, the corresponding stoichiometric coefficients of each compound consumed for biomass or virion production were multiplied by the molecular weight of the corresponding compound. The amount of ATP hydrolyzed to ADP as part of the biomass or viral replication reaction (29.25 mmol/g$_{DW}$/h for biomass, 30.62 mmol/g$_{DW}$/h for

viral replication) was subsequently divided by this weight to obtain the amount of ATP required for production of one gram of biomass or virions.

We derived a shortlist of candidate targets by subsetting the list to those enzymes listed as tier-1 targets at least once for each data set and reported as interaction partners of the proteome of at least two of the three viruses (39 candidates). The frequency at which each enzyme was listed as a tier-1 target is provided in Supplementary Data 8. Further information from the literature, CRISPR-Cas studies[32–34], and the availability of suitable inhibitors were additionally used to identify candidate targets suitable for experimental testing. To identify the gene's relevance in other viral diseases, we searched the database of protein, genetic, and chemical interactions (BioGRID)[26] and the molecular INTeraction database[104]. To identify known drugs or inhibitors, we searched the Drug Gene Interaction database[105], the DrugBank database[106], and the GeneCards suite[107]. Based on this information, we selected four enzyme complexes covering eleven of the 39 gene targets for which inhibitors were available for experimental testing.

## Computational analysis—statistics and reproducibility
Statistical tests were performed with R version 4.1.3 if not indicated otherwise. The individual statistical tests are indicated for each case in which $p$ values are reported. For all statistical tests the corresponding functions from the R base package were used. In the case of multiple tests, false discovery rate control using the Benjamini and Hochberg procedure[108] implemented in the p.adjust function of R was used. For plotting, ggplot2 was used. For number and definition of replicates for experimental approaches, please see the corresponding "Methods" sections.

## Experimental approaches—cell culture
A549 cells (human alveolar basal epithelial adenocarcinoma) were maintained at 37 °C with 5% $CO_2$ in RPMI 1640 Medium containing 10% ($v/v$) inactivated fetal calf serum (FCS) and 100 µg/mL penicillin-streptomycin. A549 cells were kindly provided by Prof. Stefan Pöhlmann and Dr. Markus Hoffmann, German Primate Center Göttingen, Germany.

Calu-3 cells (human lung adenocarcinoma cells) and HEp-2 cells (human epidermoid cancer cells) were maintained at 37 °C with 5% $CO_2$ in Dulbecco's Modified Eagle Medium (DMEM) containing 10% FCS, GlutaMax, and 100 µg/mL penicillin-streptomycin. Calu-3 cells were received from Prof. Oliver Planz, University Tübingen, Germany. HEp-2 cells were received from Dr. Katharina Rox, HZI, Hannover, Germany.

CaCo-2 (human colorectal adenocarcinoma cells) and Huh7.5 cells (human hepatocellular carcinoma cells) were maintained at 37 °C with 5% $CO_2$ in DMEM containing 10% FCS, GlutaMax, 1% ($v/v$) nonessential amino acids and 100 µg/mL penicillin-streptomycin. CaCo-2 cells were received from ATCC (Cat# HTB-37). Huh7.5 cells were kindly provided by Prof. Ralf Bartenschlager (Department of Infectious Diseases, Molecular Virology, University Heidelberg, Germany).

Huh7-Lunet-T7 RC cells (human hepatocellular carcinoma cells expressing dengue reporter construct) were maintained at 37 °C with 5% $CO_2$ in DMEM containing 10% FCS, GlutaMax, 1% ($v/v$) nonessential amino acids, 100 µg/mL penicillin-streptomycin, 1 µg/ml Puromycin and 5 µg/ml Zeocin. Huh-Lunet-T7 RC cells were kindly provided by Prof. Ralf Bartenschlager (Department of Infectious Diseases, Molecular Virology, University Heidelberg, Germany).

## Experimental approaches—viruses
Two different SARS-CoV-2 strains were used in this study and experiments involving replication-competent SARS-CoV-2 were conducted in a BSL3 laboratory. First is the recombinant SARS-CoV-2 clone expressing mNeonGreen icSARS-CoV-2-mNG[109]. It was obtained from the World Reference Center of Emerging Viruses and Arboviruses at the University of Texas Medical Branch. For virus production, CaCo-2 cells were infected; 48 h post-infection (hpi), the supernatant was collected, centrifuged, and stored at −80 °C. Second is a clinical SARS-CoV-2 isolate that belongs to the B.1.1.529 (Omicron) BA.1 lineage. It was isolated from a PCR-positive patient by a throat swab at the Institute for Medical Virology and

Epidemiology of Viral Diseases, University Hospital Tübingen, Germany. Briefly, 200 µL of patient material was used to inoculate CaCo-2 cells in a six-well plate (150,000/well). At 48 hpi, the supernatant was collected, centrifuged, and stored at −80 °C. After two consecutive passages, samples were prepared for next-generation sequencing, and the correct SARS-CoV-2 lineage was determined. The MOI was determined for both viruses by titration with serial dilutions. The number of infectious virus particles per mL was calculated as the (MOI × cell number)/(infection volume), where MOI = −ln(1 − infection rate).

The recombinant respiratory syncytial virus (RSV rA2-eGFP) was kindly provided by Jun.-Prof. Konstantin Sparrer (Institute of Molecular Virology, University Hospital Ulm, Germany) and Assoc. Prof. Michael N. Teng (University of South Florida Morsani College of Medicine, Tampa, Florida, USA)[110]. HEp-2 cells were infected, and the cells and the supernatant were harvested at 72 hpi, sonicated for 10 min at 35 °C, centrifuged, and stored at −80 °C to generate rA2-eGFP stocks.

The recombinant IAV expressing GFP (IAV-GFP SC35M) was kindly provided by Jun.-Prof. Konstantin Sparrer (Institute of Molecular Virology, University Hospital Ulm, Germany) and Prof. Martin Schwemmle (Institute of Virology, University Hospital Freiburg, Germany)[111].

For the dengue studies, recombinant dengue 2 strains 16681 (Genebank Accession NC_001474) or recombinant dengue 2 strains 16681 (Genebank Accession NC_001474) containing a Renilla Luciferase Reporter was used[112–115]. The dengue genome was transcribed as viral RNA in full-length and electroporated into Huh7.5 cells or Huh7-Lunet-T7 RC cells 3 days and 5 days post electroporation, the supernatant was collected, centrifuged, and stored at −80 °C.

## Experimental approaches—compound information
Atpenin A5 (Cat# sc-202475A) and scyllo-inositol (Cat# sc-202808) were obtained from Santa Cruz Biotechnology (Dallas, Texas, USA) and dissolved in DMSO for atpenin A5 and in HPLC water for scyllo-inositol. Phenformin (Cat# P7045) and SR13800 (Cat# 5096630001) were purchased from Merck (Rahway, New Jersey, USA) and dissolved in HPLC water. Metformin (Cat# AG-CR1-3689) was acquired from AdipoGen Life Sciences (San Diego, California, USA) and dissolved in HPLC water.

## Experimental approaches—initial screening of four drug candidates against
SARS-CoV-2 CaCo-2 and Calu-3 cells were seeded into a 96-well flat bottom plate with $1 × 10^4$ (CaCo-2) or $4 × 10^4$ (Calu-3) cells per well. The next day, cells were pre-treated with phenformin, SR13800, and scyllo-inositol in concentrations of 50 µM, 10 µM, and 2 µM, and atpenin A5 in concentrations of 10 µM, 2 µM, and 0.4 µM. After 24 h, cells were infected with icSARS-CoV-2-mNG[109] at a multiplicity of infection (MOI) = 0.2 for CaCo-2 cells and at an MOI = 0.5 for Calu-3 cells or mock-infected. After 48 h post-infection (hpi), cells were fixed with 2% PFA and stained with Hoechst 33342 (Thermo Fisher Scientific, Cat# H1399) at 1 µg / mL final concentration. Images of cell nuclei, mNeonGreen, and bright field were taken with Cytation3 (Biotek, Winooski, VT, USA). Hoechst+ and mNeonGreen + cells were counted by the Gen5 software (Biotek, Winooski, VT, USA), and infection rates were calculated using the ratio of mNeonGreen +/Hoechst+ cells.

## Experimental approaches—IC50 calculation of phenformin, metformin, and atpenin A5
For $IC_{50}$ calculations with the SARS-CoV-2 strains, Calu-3 cells were seeded into a 96-well flat bottom plate with $4 × 10^4$ cells per well. The next day, cells were pre-treated with phenformin, metformin, or atpenin A5. After 24 h, cells were infected with icSARS-CoV-2-mNG at a MOI = 0.5 or with SARS-CoV-2 Omicron at a MOI = 1.1. The icSARS-CoV-2-mNG infected cells were fixed, stained, and measured as described before. The SARS-CoV-2 Omicron infected cells were fixed with 80% Acetone and blocked with 10% normal goat serum (Cell Signaling Technology, Cat# 5425). Afterwards cells were stained by Immunofluorescence with rabbit anti-SARS-CoV-2

nucleocapsid antibody (GeneTex, Cat# GTX135357, RRID:AB_2868464) 1:1,000 in PBS and with goat anti-rabbit AlexaFluor™ 594 antibody (Thermo Fisher Scientific, Cat# A-11012, RRID:AB_2534079) 1:2,000 in PBS. Cell nuclei were stained with DAPI (Merck, Cat# D9542) 1:20,000 in PBS. Images of cell nuclei, AlexaFluor™ 594+ cells, and bright field were taken with the Cytation3 (Biotek, Winooski, VT, USA), and DAPI+ and AlexaFluor™ 594+ cells were automatically counted by the Gen5 software (Biotek, Winooski, VT, USA).

For $IC_{50}$ calculations with dengue virus, phenformin or atpenin A5 were prediluted in Huh7.5 complete media and added to Huh7.5 cells 1 h before adding the virus (MOI = 1). At 24 h post-infection, a lysis buffer was added to the cells, and the cell lysates were subjected to luciferase assay and measured with Cytation3 (Biotek, Winooski, VT, USA).

For $IC_{50}$ calculation with IAV or RSV, A549 cells were pre-treated with phenformin or atpenin A5. Cells were infected with IAV or RSV 24 h post-treatment with an MOI of 0.6 (IAV), MOI of 1.4 (RSV), or mock-infected. After 24 h (IAV) or 48 h (RSV), cells were fixed with 2% PFA and stained with Hoechst 33342. The plates were measured with Cytation3 (Biotek, Winooski, VT, USA), and Hoechst+ and GFP+ cells were counted by Gen5 software (Biotek, Winooski, VT, USA).

$IC_{50}$s of all viruses were calculated as the half-maximal inhibitory dose using the GraphPad Prism 9 Software (GraphPad Software, Inc., San Diego, CA, USA, Version 9).

### Experimental approaches—CC50 calculation of phenformin and atpenin A5

$CC_{50}$ calculations were performed using three methods to measure cell viability: the MTT assay, the CellTiter-Glo® assay (CTG) from Promega (Cat# G7570), and monitoring cell viability via live cell imaging (IncuCyte®). For the MTT assay, Calu-3, Huh7.5, and A549 cells were treated with phenformin in concentrations of 6400 μM to 0.2 μM and atpenin A5 in concentrations of 640 μM to 0.15 μM in two-fold dilution for 72 h. Atpenin A5 was dissolved in DMSO. Therefore, DMSO control with 1.3% (≙ 640 μM atpenin) was added to exclude false positive cell toxicity. The positive control is 50% DMSO to inhibit cell growth. After 72 h of compound incubation, cells were washed with PBS and 10% MTT solution (Abcam, Cat# ab146345) was added in phenol red-free medium for 3 h at 37 °C. Cells were then incubated in 0.04 M HCl in isopropanol and plates were gently shaken for 10 min at room temperature. Absorption levels at 570 nm and 650 nm wavelengths were measured by a Berthold TriStar2 S Multimode Reader, and values were normalized to non-treated cells. $CC_{50}$ was calculated as the half-maximal cytotoxic dose via GraphPad Prism 9 using four-parameter nonlinear regression.

For CTG, Calu-3, Huh7.5, and A549 cells were treated with the same concentrations as listed above, and DMSO controls and positive control with 50% DMSO treated cells were added. The assay was performed in accordance with the manufacturer's instructions. A Berthold TriStar2 S Multimode Reader was used to measure luminescent signals. The data was then normalized to the non-treated control. $CC_{50}$ was calculated as the half-maximal cytotoxic dose via GraphPad Prism 9 using four-parameter nonlinear regression.

For live cell imaging, Calu-3, Huh7.5, and A549 cells were treated with phenformin in concentrations of 12800 μM to 0.2 μM and atpenin A5 in concentrations of 640 μM to 0.15 μM in two-fold dilution. Next, plates were stored in the IncuCyte® (Sartorius AG, Göttingen, Germany) at 37 °C with 5% $CO_2$. Cell confluence was measured every three to 4 h via a phase channel with a 10× objective. For analysis, a basic analyzer of the IncuCyte® S3 Software (Sartorius AG, Göttingen, Germany) was used.

### Experimental approaches—in vivo infection experiments

Infection experiments in Golden Syrian hamsters were performed in the animal facility of the Université de Lyon, VetAgro Sup, Institut Claude Bourgelat (69280 Marcy l'Etoile, France). The experimental protocol was authorized by the Institutional Ethics Committee of VetAgro Sup (CEEA 18, project number 2066) and the French Ministry (APAFIS#27797-2020100516408472), and we have complied with all relevant ethical regulations for animal use.

Sixteen 9–11 week-old female immunocompetent Golden Syrian hamsters (Janvier Labs, Le Genest-Saint-Isle, France) were randomized according to their weight in two groups of 7 and one group of 2, with each individual animal being considered an experimental unit. Animals were confirmed to be seronegative for SARS-CoV-2 for their inclusion in the study. Sample size was calculated to have at least 3 replicates on each infected group for every time-point in which lungs were collected and. Animals were housed in micro-isolator cages (maximum 4 animals per cage) in a biosafety 3 controlled environment (22 °C, 30–70% humidity, 12:12 h photoperiods, 10 air cycles/h), with *ad libitum* access to food and water. Cages belonging to each group were clearly labeled to avoid potential confounders. On day 0 (D0), the two groups of 7 animals were anesthetized with inhaled isoflurane/oxygen and infected intranasally with $10^{5,5}$ TCID50 of a SARS-CoV-2 Delta (B.1.617.2) strain in 60 μL of PBS. The group of 2 animals was mock infected with 60 μL of PBS. All animals received one 90 μL gavage per day for 5 days between D-2 and D2 containing 100 mg/kg phenformin or sterile water, as indicated in each case. Animals were weighted and monitored for clinical signs on D-3 (baseline before treatment initiation) and then daily after infection. Oropharyngeal swabs were performed daily between D1 and D6 and immediately stored at –80 °C for further total RNA extraction. A subset of animals were euthanized on D3 and D6 and their lungs were removed and homogenized in cold PBS to minimize degradation before total RNA extraction. Total RNA from oropharyngeal swabs and lung homogenates were used for nsp14 gene quantification by RT-qPCR, with operators being blinded for the nature of each group (treatment vs. vehicle).

### Reporting summary

Further information on research design is available in the Nature Portfolio Reporting Summary linked to this article.

### Data availability

All data produced in the context of this work is available in the Supplementary Data. Links to source data for the different scRNA-Seq data sets are provided in Table 2. The human metabolic models with all constraints for the blood medium and the viral replication reactions have been uploaded to the Bio-Models Database[93] (Table 3) under the following accessions: Dengue, https://www.ebi.ac.uk/biomodels/MODEL2311070006; Influenza, https://www.ebi.ac.uk/biomodels/MODEL2311070002; SARS-CoV-2 BALF1, https://www.ebi.ac.uk/biomodels/MODEL2311070001; SARS-CoV-2 BALF2, https://www.ebi.ac.uk/biomodels/MODEL2311070007; SARS-CoV-2 CALU3, https://www.ebi.ac.uk/biomodels/MODEL2311070005; SARS-CoV-2 scH1299, https://www.ebi.ac.uk/biomodels/MODEL2311070004. Figure source data can be found in Supplementary Data 3.

### Code availability

Scripts to reproduce the analyses are available via GitHub in the repository https://github.com/draeger-lab/R-DRUGS (Zenodo-Archive: https://doi.org/10.5281/zenodo.15103806[116]). The program that produced the hierarchical model versions can be found in this repository https://github.com/draeger-lab/ModelEditingTools as the `TissueModelExtractor` (Zenodo Archive: https://doi.org/10.5281/zenodo.15103819[117]).

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

## Acknowledgements

This work was supported by the BMBF-funded de.NBI Cloud within the German Network for Bioinformatics Infrastructure (de.NBI) (031A532B, 031A533A, 031A533B, 031A534A, 031A535A, 031A537A, 031A537B, 031A537C, 031A537D, 031A538A). C.K. acknowledges support by the Deutsche Forschungsgemeinschaft (DFG, German Research Foundation) within the cluster of excellence "Precision medicine in chronic inflammation" (DFG support code EXC2167), the collaborative research center "Metaorganisms" (DFG support code CRC1182), the research group miTarget (DFG support code FOR5042) and the German Ministry for Education and Research in the frame of iTREAT (BMBF support code 01ZX1902A). A.D. was further supported within the project EXC-2124/1-05.037_0 under Germany's Excellence Strategy–EXC 2124 390838134 (Cluster of Excellence "Controlling Microbes to Fight Infections") and by the German Center for Infection Research (DZIF) within the Deutsche Zentren der Gesundheitsforschung (funded by the German Ministry of Education and Research), grant no. 8020708709. M.S. and R.J. are funded by the German Center for Infection Research (DZIF), grant no. FF 01.907. M.H. and M.B. were fellows of an M.D. stipend granted by the German Center for Infection Research (DZIF) under grant no. TI 07.003. We are grateful to the personnel of the Institut Claude Bourgelat for their expertise and technical support with animal studies. The graphical abstract was created with BioRender.com.

## Author contributions

A.R., A.D., and C.K. devised the computational component of the study. M.S. conceived and devised the experimental part of the study. M.R.-C., S.M.B., M.P., M.D., M.B., and A.P. conceived and devised the animal experiments. A.R., M.H., J.J.-S., A.P., A.D., M.S., and C.K. wrote the initial manuscript draft. A.R., J.J.-S., L.B., C.K., G.M., N.L., and F.C. analyzed the computational data and contributed to identification of antiviral compounds. M.H., M.B., J.D., R.J., V.D., C.M., M.R.-C., A.P., S.M.B., M.P., M.D., M.B., and M.S. conducted all virus infection experiments and measurements of cellular metabolic activity and toxicity and analyzed the data. All authors contributed to manuscript editing and approved the final paper.

## Funding

## Competing interests

The authors declare the following competing interests: A.R., M.H., L.B., J.J.S., A.D., M.S., and C.K. have submitted a patent application for using phenformin and atpenin A5 as broad-spectrum antivirals. All other authors declare no competing interests.

## Additional information

[1]Computational Systems Biology of Infections and Antimicrobial-Resistant Pathogens, Institute for Bioinformatics and Medical Informatics (IBMI), Eberhard Karl University of Tübingen, Tübingen, Germany. [2]Institute for Medical Virology and Epidemiology of Viral Diseases, University Hospital Tübingen, Tübingen, Germany. [3]German Center for Infection Research (DZIF), partner site, Tübingen, Germany. [4]Research Group Medical Systems Biology, Institute of Experimental Medicine, Christian-Albrechts-University Kiel & University Hospital Schleswig Holstein, Kiel, Germany. [5]CIRI, Centre International de Recherche en Infectiologie (Team VirPath), Université de Lyon, INSERM U1111, Université Claude Bernard Lyon 1, CNRS, UMR5308, ENS de Lyon, Lyon, France. [6]VirNext, Faculté de Médecine RTH Laennec, Université Claude Bernard Lyon 1, Université de Lyon, Lyon, France. [7]International Research Laboratory RESPIVIR France—Canada, Centre Hospitalier Universitaire de Québec - Université Laval, Québec, Canada, Centre International de Recherche en Infectiologie, Faculté de Médecine RTH Laennec, Université Claude Bernard Lyon 1, Université de Lyon, INSERM, CNRS, ENS de Lyon, Lyon, France. [8]Division of Infectious Diseases and Immune Defense, The Walter and Eliza Hall Institute of Medical Research, Melbourne, VIC, Australia. [9]Department of Computer Science, Eberhard Karl University of Tübingen, Tübingen, Germany. [10]Cluster of Excellence 'Controlling Microbes to Fight Infections', Eberhard Karl University of Tübingen, Tübingen, Germany. [11]Cologne Excellence Cluster for Cellular Stress Responses in Aging-Associated Diseases (CECAD), University of Cologne, Cologne, Germany. [12]Data Analytics and Bioinformatics Research Group, Institute of Computer Science, Martin Luther University Halle-Wittenberg, Halle (Saale), Germany. [13]Present address: Wolfson Wohl Cancer Research Centre, School of Cancer Sciences, University of Glasgow, Glasgow, UK. [14]These authors contributed equally: Alina Renz, Mirjam Hohner. [15]These authors jointly supervised this work: Andreas Dräger, Michael Schindler, Christoph Kaleta. ✉e-mail: andreas.draeger@informatik.uni-halle.de; michael.schindler@med.uni-tuebingen.de; c.kaleta@iem.uni-kiel.de

