## [Transparent Peer Review file · Communications Biology]

Metabolic modeling elucidates phenformin and atpenin A5 as broad-spectrum antiviral drugs against RNA viruses

Corresponding Author: Professor Christoph Kaleta

Version 0:

Reviewer comments:

Reviewer #1

(Remarks to the Author)

The manuscript by Renz et al. develops and tests the idea that targeting host molecules key to viral replication might be an efficient and valuable strategy for controlling pathogen replication. The authors developed and implemented a multistep search strategy that included the computational analysis of mammalian cell metabolic models, the viral biomass function, RNA-seq data, and score functions to predict the infection models, the candidate metabolic pathways, and their contribution/impact on viral replication. SARS-CoV-2, Dengue, and Influenza A were selected as target viruses.

The candidate metabolic pathways, and proteins thereof, were ranked and grouped according to their essentiality for cell viability. Inhibitors against a small subset of proteins/complexes (Complex I and II from the mitochondrial electron transport chain: ETC; the Soluble carrier family 16 member 1 gene: monocarboxylic acid transporter; and the CDP-diacylglycerol-inositol 3-phosphatidyltransferase) were tested in vitro for their capacity to inhibit the replication of SARS-CoV-2. Phenformin (Complex I inhibitor) and Atpenin 5 (complex II inhibitor) displayed anti-SARS-CoV-2 activity and were further tested against a different SARS-CoV-2 variant (Omicron), Dengue virus, the Respiratory Syncytial Virus and Influenza A Virus. Atpenin 5 proved active against all single-stranded RNA virus assayed whereas Phenformin showed discrete activity against Dengue virus, in addition to SARS-CoV-2. The cytotoxicity of both compounds against the different host cells was assessed to determine their antiviral selectivity, which was significantly high when cell proliferation/viability was measured by a non-metabolic assay (cell confluency estimated by automated microscopy observation). Based on drug safety issues, Phenformin was selected for a single dose (100 mg/Kg per oral) in vivo therapeutic efficacy study performed on a Hamster model of SARS-CoV-2 infection. The compound proved effective in lowering viral titers in biological samples from the oropharyngeal tract but fully ineffective in controlling viral infection and replication in lung tissue.

Overall, the experimental design appears appropriate for the questions set by the authors. However, some conclusions are over-ranked.

Here is a list of issues (note: it is not ordered according to relevance) that the authors may consider during the revision:

- The title is misleading because it suggests that Phenformin and Atpenin A5 have activity against "all viruses", which is not correct. On the one hand, the first proved inactive against RSV and IAV, and on the other hand, the compounds were only tested against a specific type of virus: single-stranded RNA viruses. Consider modifying the title to fits the findings of the study.
- Figure 1 A: the final step presented in scheme 1 suggests that a virtual drug screening was performed for the protein candidates affecting viral replication. By reading the corresponding Results and Mat&Met sections, it seems that the inhibitors tested in this study were selected empirically and not applying computational methods. If this interpretation is correct, then modify the scheme; if not, then describe in Mat&Met the method applied to identify and select the compounds tested.
- at several parts of the text it is used the term "interaction partners" or "interacting proteins" from a biophysical standpoint. Do the authors refer to "physical interaction partners" or to "functional interaction partners". The first implies a physical interaction/contact between the host proteins and some viral components, whereas the last assumes a functional dependency of the virus on specific host proteins. Please clarify this.

- during viral infection and depending on cell type, some proteins are up- or down-regulated. This is the consequence of a complex host-cell/pathogen interaction. Has the search for the "interaction proteins" of the viral proteome identified in this study been unbiased regarding the expression level of the host proteins?

- line 218: it is stated that "254 enzymes" belonging to Tier-1 targets were shared among all datasets. This figure does not match the absolute values for Tier-1 targets reported in Figure 2, except that there are redundant enzymatic entities among the "254 enzymes" or that the Tier-1 targets include protein complexes made of several enzymatic subunits. Please clarify this.

- line 279-281: it would be worth briefly mentioning the subcellular location of the SLC16A1 transporter (plasma membrane, mitochondrial membrane, or ubiquitous).

- line 278 and 290-292: Phenformin has recently been tested against SARS-CoV-2 (Grau-Expósito et al. PLoS Pathogens, DOI 10.1371/journal.ppat.1010171). The compound inhibited the entry of SARS-CoV-2-like particles in HTL but not in Vero cells. Atpenin 5 has recently been reported to inhibit SARS-CoV-2 MPro but lacked antiviral activity when administered to cells along with the virus (Prada et al. 2023 Frontiers Drug Discov, DOI: 10.3389/fddsv.2022.1082065). It may be worth discussing/contrasting this result with the ones generated in your study.

One important point is that, at variance with the studies quoted above, in the present research, the SARS-CoV-2 infection assays were performed in cells pre-treated with compounds 24 h pre-infection. The rationale and consequence of this metabolic pre-conditioning of the cells should be discussed. Cells with a previously impaired energetic metabolism will not offer optimal conditions for virus internalization (which is an active process) and replication. Addressing this point is very relevant because, though missed by the authors, may have influenced the selection of the in vivo therapeutic approach (prophylactic rather than therapeutic) with Phenformin (see comments below).

If possible, the SARS-CoV-2 infection assay on Calu-3 cells should be repeated and treatment with Phenformin and Atpenin 5 initiated concomitantly with virus infection. This may help to discriminate at which step of the virus cycle these drugs interfere the most.

- line 310: please comment on why Scillo-inositol promoted rather than inhibited SARS-CoV-2 replication.

- lines 328 and 329: The inactivity of phenformin against RSV and IAV should be further discussed in the Discussion section. This result suggests a virus/host-cell-specific action of phenformin and, hence, restricts the emphasis on the "broad spectrum" label assigned to this drug.

- the capacity of Phenformin to lower viral load in the upper respiratory tract is promising. However, there are two key factors that, in principle, raise some doubts about its prospective as a "therapeutic" anti-SARS-CoV-2 drug candidate. The therapeutic regimen was initiated 72 hours in advance of infection and failed to control virus replication in a physiologically key target tissue: the lungs. As such, Phenformin (or analogs: Metformin) may be effective as a prophylactic drug transiently applied to risk groups (e.g. health personnel, patients with comorbidities compatible with Metformin or Phenformin administration). The authors may discuss more objectively these points.

- Concerning the comment above, and taking into account the massive information about the interconnected metabolic pathways relevant for viral replication obtained in this study, it would have been interesting that the authors propose, if feasible, which of them (from Tears-1 targets) may be subject of simultaneous inhibition to achieve synergistic effect.

- Please indicate the SARS-CoV-2 variant the virus used for the screening (the mNeon Green...) belongs to.

Reviewer #2

(Remarks to the Author)
General Comments

The authors use metabolic network modeling to identify enzymes that, when inhibited, could prevent viral replication by diminishing the rates of energy generation or the production of viral biomass precursors, without adversely affecting the host's cellular metabolism. They include various viruses in their analysis and find common targets, thus making the study relevant for pandemic preparedness. Despite starting with sparse single-cell data, the modeling pipeline successfully highlights differences in viral replication potential between cells from infected and healthy individuals, and predicts targets that are known to interact with viral proteomes. It is particularly interesting and insightful that cells that are from patients but are not yet infected are predicted to be metabolically primed to support viral replication. The authors then focus on a subset of target proteins that have been experimentally verified to interact with these viruses. Two available drugs that target proteins from this group are experimentally validated to have significant potential. It is not feasible to validate systems-level modeling predictions in a high throughput manner. Also, since the choice of targets and drugs for validation relied on their previously known interactions and historical data on their relevance and success potential, it is difficult to fully assess this modeling pipeline's utility for therapeutic applications. Therefore, it is critical that the modeling approach is free of concerns (please see my major comments) and delivered clearly (please see my minor comments).

Major Comments

In principle, halting the replication of a virus (or the proliferation of a tumor) by suppressing biomass or energy production is straightforward, but maintaining patient health during this process is a significant challenge. The modeling predictions of this study are based on the biomass reactions of both the virus and the human host. Thus, a critical part of the analysis involves optimizing potential targets, defining an ideal target as one that maximally reduces viral biomass production while minimally affecting human biomass production. The Methods section specifies two thresholds for defining these targets: a maximum 50% reduction in viral biomass and a minimal 20% reduction in human biomass (I am omitting the details on tier-1 and tier-2 targets for simplicity). Despite the importance of this optimization, the rationale for choosing these particular thresholds and the workings of this strategy are unclear, so I have several questions:

Given the shared components and ATP requirements in the biomass reactions of both the virus and humans, what makes viral replication so vulnerable when human biomass production is not?

Why do enzymes involved in the TCA cycle and electron transport chain appear non-essential in the human metabolic model, not contributing to significant reductions (>>20%) in human biomass production?

How sensitive is the outcome to the thresholds used? Could these thresholds be overfitted to known druggable targets in the TCA cycle and ETC?

How do the predicted common targets compare with straightforward predictions from traditional flux balance analysis using the biomass objective functions without integrating any data?

For this work to be convincing, these aspects must be thoroughly addressed, possibly requiring additional analyses and figures.

Minor Comments

* Uptake rates (Table S8): At a first glance, it makes no sense to directly use metabolite concentrations (units of mmol/L) as exchange flux rates (units of mmol/g.h). But, I understand that this is considered as a first approximation that is acceptable in a semi-quantitative modeling setting (FASTSCORE integration). And I agree that it may be sufficient for this case. But this point deserves some clarification and more discussion for the modeling audience.

* Lines 653-655: "Gene level counts were translated into transcripts per million (TPM) values through normalization according to human ENSEMBL gene lengths".
Single cell data is based typically on 3' or 5' end reads, not full-length reads, and therefore gene length is irrelevant in most cases. I checked only one of the datasets that the authors used (Liao et al., 2020; data can be traced back to <https://www.ncbi.nlm.nih.gov/geo/query/acc.cgi?acc=GSM4339769>), and got the impression that it is based on 5'-end reads. Please clarify this matter regarding data normalization.

* Discussion: A big part of the discussion is repeating the results (including but not limited to the second and third paragraph). Please consider revising to focus more sharply on critical topics needing further exploration and clarification.

* Fig. 2, legend: The opening is unclear and convoluted. Please rephrase.

* Line 311: I believe the referral to Fig 4B is a typo and it should be 4C.

* Line 329: The manuscript claims no severe cytotoxicity was observed for the compounds tested, yet some data points show viable cell counts below 50%. Is this level not considered toxic?

* Fig. 4, legend: mNG is used in abbreviated and non-abbreviated form in different places.

* Line 421: What is ARN? (Possibly a typo for RNA, but this is repeated within the same paragraph as well as in Fig 7)

* Line 463: The assertion that virus-induced cellular senescence contributes to a pro-inflammatory microenvironment is counterintuitive if senescence is supposed to coincide with increased metabolic activity necessary for viral replication. This paradox needs clarification.

* Line 599, Table S6: This reference should likely be to "Table S7." Additionally, the terms "VBOF" are used to refer both to viral biomass objective functions and viral replication reactions, which could confuse readers. Please eliminate the latter term to avoid confusion.

Version 1:

Reviewer comments:

Reviewer #1

(Remarks to the Author)

The authors considered the most relevant suggestions (including new experimental information) made in the revision and satisfactorily answered all the questions raised.

I have no more concerns and recommend its publication.

Reviewer #2

(Remarks to the Author)

The authors have properly addressed my major comment on sensitivity and all minor comments. However, concerns about the source of mechanistic predictions (specifically, how the model predicts that the perturbation of an enzyme significantly impacts viral replication a lot more than human biomass production) remain inadequately addressed. The responses provided in the rebuttal are vague, and no corresponding modifications have been made to the manuscript for readers to evaluate these claims.

I also see that some of the other comments could not be perfectly addressed because redoing the computational analysis is challenging. This difficulty underscores the importance of clearly elucidating the inner workings of the predictive algorithms. If executing this modeling task is challenging even for the authors, who have the pipeline readily available, how can a potential user of the approach, like myself, confidently launch a project without being convinced of its robustness? As I mentioned previously, validating only two predictions out of hundreds does not sufficiently demonstrate the utility of the model, especially when those targets were selected based on other sources of evidence as well. The only way to build confidence in a model of this scale is through internal validation, which would explain the reasoning behind its predictions.

Since some of my comments appear to have been misunderstood or misinterpreted, I will now explicitly outline what a satisfactory response would look like, offering an example to avoid further ambiguity.

To clarify the source of modeling predictions, I would suggest the following:

- *Take two example predictions, possibly from Fig. 3B (a blue and a red reaction). Show that regular FBA using a generic model would fail to predict these outcomes (if it doesn't, an alternative example can be chosen to replace one of these).
- *Demonstrate that something specific to the integrated models leads to these predictions. For instance, it might be that the TCA cycle is relatively less important in the target cells, as indicated to the integration algorithm via gene expression data.
- *Next, explain how the energy requirements of these cells are met (e.g., through glycolysis, beta oxidation, or possibly reduced overall energy demands as predicted by the integration).
- *Ideally (not necessarily), use this analysis to connect the modeling predictions to experimental results, such as those summarized in Fig. 6.

I must stress that this is just a hypothetical example to clarify how the missing elements could be addressed; the actual implementation will depend on the data and findings at hand. For example, the example chosen might be about a biomass precursor that the virus needs but the human cell does not. Nonetheless, this level of detail is essential to understand and trust the mechanistic basis of the predictions. While showing subsystem enrichments of predictions is a helpful start (Fig. 3A), it does little to explain the mechanisms driving those predictions.

Version 2:

Reviewer comments:

Reviewer #1

(Remarks to the Author)

Editor comment: I asked this reviewer to comment on points #3, #4, and #6 to reviewer #2 in the rebuttal letter.

Their arguments on point 3 are consistent with the parametrization and selection filter for proposing the candidate host genes contributing to viral replication. Eventually, to address the reviewer's concern, they may mention any potential caveat of the search algorithm strategy used to notify other researchers about some limitations the method may have or to open new avenues for improvement.

Regarding point 4, the authors have provided direct evidence that, if not combined with transcriptomic data, the FBA prediction model fails to pick up physiologically relevant candidate genes. Although they prefer not to comment on the results of these control experiment, to attend the pertinent request of the reviewer, they can briefly mention in the manuscript (Mat&Met or Result section, when discussing how the search strategy was validated) something like this: "Applying the used cutoffs (<50% maximal viral replication rate, >80% maximal biomass production rate) on the human metabolic model with our predefined diet without the mapping of transcriptomic data, did not yield any tier-1 targets. From the two knockout targets identified as candidates to affect viral replication (asparagine synthetase: ASNS, and cytidine/uridine monophosphate kinase 1: CMPK1), ASNS knockout reduces maximal biomass production rate to 57%, while CMPK1 knockout is lethal. This result highlights the relevance of combining generic FBA modeling with gene expression data."

Concerning point 6, I agree with the authors that it is not necessary to explain the cells' energy requirements, which, independent of viral infection, will depend on nutrient/substrate availability, cell cycle stage, and other less important factors.

Clearly, viral replication demands a lot of ATP and the most efficient energetic metabolism to produce it is the TCA-cycle coupled to the mitochondrial electron-transport chain.

To support this statement, the authors should consider that nucleic acid biosynthesis is an energetically costly process; at least one mol ATP is consumed for each base ligated to the nucleic acid chain (RNA in the case of SARS-CoV2). Therefore, any pharmacological strategy reducing cellular ATP will efficiently affect the replication of non-integrative and highly replicating viruses.

Reviewer #2

(Remarks to the Author)

The authors have made some modifications that further clarify aspects of the viral component of modeling predictions. However, the key question remains unanswered: why does the integrated model predict a lower impact on human biomass production than on viral biomass production for tier 1 enzymes and some others, whereas regular FBA with the viral biomass reaction does not? This lack of mechanistic understanding undermines confidence in the predictions, which was the main point of my original comments.

In the itemized rebuttal, the most relevant response would be that under Comment #5. However, that response directs me to Comment #3, where there is no related answer. Even if #3 was referenced by mistake, I cannot find a clear response elsewhere. While the authors do not explicitly state it, the implication seems to be that the mechanistic basis of predictions does not matter because the model is validated by data. Even assuming the validation is convincing, this approach effectively treats a mechanistic model as a partial black box (though I appreciate the effort to clarify how the model calculates viral biomass production). The value of mechanistic models, however, extends beyond predictive accuracy; they are meant to provide biological insight. If the mechanistic basis is ignored, what remains is little more than a method for filtering candidate targets from 84 to 39, which raises the question if this reduction alone worth the effort, as the authors indicated themselves. Furthermore, without an understanding of the mechanistic basis, the possibility of overfitting cannot be ruled out. While I appreciate the previous sensitivity analysis addressing primary thresholds for predictions, complex models of this nature contain many potential sources of arbitrariness; including threshold choices when processing the experimental data, constraint settings, and even model selection itself. The most effective way to dispel such skepticism is to demonstrate how unexpected predictions arise.

Below are specific comments in a third attempt to clarify this key issue:

1) Regular FBA predictions remain unmentioned in the manuscript: As in the last round, the rebuttal states (Response to Comment #4) that including this analysis in the manuscript would be distracting. I disagree. Whether regular FBA without integration can make similar predictions is a fundamental question that will concern many experts. I strongly recommend including a concise analysis of this, ideally in the Results or, even in the Discussion. Furthermore, some Discussion text remains redundant with the Results (especially most of the second paragraph), which could be trimmed to make room for this analysis.

2) What drives the difference in predictions between integrated models and regular FBA? Does the integrated model predict lower human biomass production than the generic model? If so, which pathways are responsible for this? Or does the difference stem from viral biomass production capacity? (Although, given that biomass production rates can only decrease upon integration, this seems unlikely.) Addressing these questions in the Results would greatly enhance clarity. As it stands, readers cannot even ask these questions because the necessary context is missing (please see #1 above).

3) The relationship between human and viral biomass predictions: Should human biomass production be maximized in the same cells where viral replication is most vulnerable to perturbation? While the goal is to inhibit viral replication in infected cells, human metabolism should remain unaffected in all tissues. Is this correct? If so, the rationale for maximizing both human and viral biomass in the same cells during target prediction should be explicitly stated.

4) Stronger data presentation for model validation is needed: The best argument for model validation is the enrichment of experimentally supported protein interactions in tier 1 predictions (Response to Comment #3). However, the current data presentation does not sufficiently support this point.

* Table S5 is well-organized but limited to 39 common targets. Based on its annotations and numbers, the enrichment of virus-host interactions seems largely driven by electron transport chain (ETC) enzymes, which are likely interdependent in the flux space. So, based on what we can see, it is possible that the seemingly low p-value ($9.1E-12$) is driven by multiple targets originating from a single mechanistic effect.

* Tables S2 and S3 could have shown if this is the case for the data from where the enrichment score comes, but these tables are not useful in their current format. To enable evaluation, I strongly recommend providing a master table of all interactions and tier-1/2 predictions in a clearer format similar to Table S5.

* Does the ETC alone drive enrichment? Even after providing the annotation tables, clarifying whether enrichment is dominated by ETC-related enzymes or another lumped factor would significantly improve the interpretation of the findings.

These clarifications would enhance the manuscript's transparency and scientific rigor. I appreciate the improvements made so far and hope this feedback will help address the remaining concerns.

Reviewer #1

(our responses in italics in blue, changes pertaining to specific reviewer comments are also indicated in the version of the manuscript containing tracked changes, line numbers refer to the document containing tracked changes)

The manuscript by Renz et al. develops and tests the idea that targeting host molecules key to viral replication might be an efficient and valuable strategy for controlling pathogen replication. The authors developed and implemented a multistep search strategy that included the computational analysis of mammalian cell metabolic models, the viral biomass function, RNA-seq data, and score functions to predict the infection models, the candidate metabolic pathways, and their contribution/impact on viral replication. SARS-CoV-2, Dengue, and Influenza A were selected as target viruses.

The candidate metabolic pathways, and proteins thereof, were ranked and grouped according to their essentiality for cell viability. Inhibitors against a small subset of proteins/complexes (Complex I and II from the mitochondrial electron transport chain: ETC; the Soluble carrier family 16 member 1 gene: monocarboxylic acid transporter; and the CDP-diacylglycerol-inositol 3-phosphatidyltransferase) were tested in vitro for their capacity to inhibit the replication of SARS-CoV-2. Phenformin (Complex I inhibitor) and Atpenin 5 (complex II inhibitor) displayed anti-SARS-CoV-2 activity and were further tested against a different SARS-CoV-2 variant (Omicron), Dengue virus, the Respiratory Syncytial Virus and Influenza A Virus. Atpenin 5 proved active against all single-stranded RNA virus assayed whereas Phenformin showed discrete activity against Dengue virus, in addition to SARS-CoV-2. The cytotoxicity of both compounds against the different host cells was assessed to determine their antiviral selectivity, which was significantly high when cell proliferation/viability was measured by a non-metabolic assay (cell confluency estimated by automated microscopy observation). Based on drug safety issues, Phenformin was selected for a single dose (100 mg/Kg per oral) in vivo therapeutic efficacy study performed on a Hamster model of SARS-CoV-2 infection. The compound proved effective in lowering viral titers in biological samples from the oropharyngeal tract but fully ineffective in controlling viral infection and replication in lung tissue.

Overall, the experimental design appears appropriate for the questions set by the authors. However, some conclusions are over-ranked.

Here is a list of issues (note: it is not ordered according to relevance) that the authors may consider during the revision:

1) The title is misleading because it suggests that Phenformin and Atpenin A5 have activity against "all viruses", which is not correct. On the one hand, the first proved inactive against RSV and IAV, and on the other hand, the compounds were only tested against a specific type of virus: single-stranded RNA viruses. Consider modifying the title to fits the findings of the study.

Indeed, we have tested our approach only on single-stranded RNA viruses, which account for a considerable fraction of reported viral infections in humans and are considered to have the most substantial potential for pandemic and public health concerns. Since previous work has discussed a particular difference between the metabolic requirements of RNA versus

DNA viruses, we have modified the title to “Metabolic modeling elucidates phenformin and atpenin A5 as broad-spectrum antiviral drugs against RNA viruses”.

2) Figure 1 A: the final step presented in scheme 1 suggests that a virtual drug screening was performed for the protein candidates affecting viral replication. By reading the corresponding Results and Mat&Met sections, it seems that the inhibitors tested in this study were selected empirically and not applying computational methods. If this interpretation is correct, then modify the scheme; if not, then describe in Mat&Met the method applied to identify and select the compounds tested.

We thank the reviewer for this comment. Indeed, most of the screening procedure was computationally. Thus, we started with 1,681 potential gene targets (accounted for in the model), which were reduced to 254 targets based on the prediction of tier-1 targets using metabolic modeling. Those were further reduced to 39 gene targets belonging to 22 protein complexes based on the requirement that they interacted with one of the proteins of at least two of the three considered viruses. Among those protein complexes, we then selected four for experimental testing (accounting for 11 gene targets) also based on the availability of suitable inhibitors. Thus, only the last step involved an empirical selection of candidates, which is quite a usual approach since each compound tested requires considerable effort. Accordingly, we added the following statement in the methods (l. 818 - 820):

“Based on this information, we selected four enzyme complexes covering eleven of the 39 gene targets for which inhibitors were available for experimental testing.”

3) At several parts of the text it is used the term "interaction partners" or "interacting proteins" from a biophysical standpoint. Do the authors refer to "physical interaction partners" or to "functional interaction partners". The first implies a physical interaction/contact between the host proteins and some viral components, whereas the last assumes a functional dependency of the virus on specific host proteins. Please clarify this.

We refer to physical interaction partners. We have tried to clarify this better in the text, but we have not replaced all occurrences of “interaction partners” with “physical interaction partners” (e.g. l. 199-205) since we felt that this somewhat reduced the clarity of the text.

4) during viral infection and depending on cell type, some proteins are up- or down-regulated. This is the consequence of a complex host-cell/pathogen interaction. Has the search for the "interaction proteins" of the viral proteome identified in this study been unbiased regarding the expression level of the host proteins?

We thank the reviewer for this comment. Our approach is somewhat biased towards highly expressed proteins due to the procedure for mapping transcriptomic data to metabolic networks. This procedure, in principle, favors proteins that are highly expressed for inclusion into the metabolic network. While this effect is somewhat reduced by using StanDep preprocessing of transcriptomic data that includes lowly expressed enzymes if deemed housekeeping enzymes, a bias still persists. Please note, however, that if a lowly expressed enzyme is essential for viral replication, it will be included in the metabolic network since we enforce the availability of the viral replication reaction.

We now acknowledge this in the discussion section of our manuscript (l. 624 - 627):

“Also, the mapping approach for transcriptomic data is based on the inclusion of highly transcribed genes and therefore is biased towards more abundant proteins hence potentially missing lowly abundant proteins relevant to viral replication.”

5) line 218: it is stated that "254 enzymes" belonging to Tier-1 targets were shared among all datasets. This figure does not match the absolute values for Tier-1 targets reported in Figure 2, except that there are redundant enzymatic entities among the "254 enzymes" or that the Tier-1 targets include protein complexes made of several enzymatic subunits. Please clarify this.

We thank the reviewer for this comment. Indeed, in Fig. 2 we only show Tier-1 targets occurring in at least 5% of all cells of a dataset as we indicate in the legend of Fig. 2 (“For each dataset, the fraction of genes that are experimentally-determined interaction partners of the viral proteomes among tier-1 and tier-2 targets occurring in at least 5% of all cells of a dataset with tier-1 targets was determined.”). For the selection of shared tier-1 targets, we require that an enzyme is identified as a tier-1 target in at least one cell across all datasets/viruses. We now state this clearer in the methods section (l. 793-796):

“To identify broad-spectrum antiviral targets, we required that the corresponding enzyme was identified as a tier-1 target in at least one cell in each dataset, yielding a total of 254 shared tier-1 targets across all datasets/viruses.”

and the results (l. 216 - 219):

“To this end, we collected all predicted tier-1 targets across all cells from the individual datasets and identified 254 enzymes that occurred as shared tier-1 targets in at least one cell across all datasets (Supplementary Table S3 and Supplementary Data S1)”

6) line 279-281: it would be worth briefly mentioning the subcellular location of the SLC16A1 transporter (plasma membrane, mitochondrial membrane, or ubiquitous).

We added this information (l. 285 - 286, addition in red):

*“The solute carrier family 16 member 1 gene (SLC16A1), a monocarboxylic acid-transporter that transports compounds such as pyruvate, lactate, branched-chain amino acids, and ketone bodies **across the cell membrane**”*

7) line 278 and 290-292: Phenformin has recently been tested against SARS-CoV-2 (Grau-Expósito et al. PLoS Pathogens, DOI 10.1371/journal.ppat.1010171). the compound inhibited the entry of SARS-CoV-2-like particles in HTL but not in Vero cells. Atpenin 5 has recently been reported to inhibit SARS-CoV-2 MPro but lacked antiviral activity when administered to cells along with the virus (Prada et al. 2023 Frontiers Drug Discov, DOI: 10.3389/fddsv.2022.1082065). It may be worth discussing/contrasting this result with the ones generated in your study.

One important point is that, at variance with the studies quoted above, in the present research, the SARS-CoV-2 infection assays were performed in cells pre-treated with

compounds 24 h pre-infection. The rationale and consequence of this metabolic pre-conditioning of the cells should be discussed. Cells with a previously impaired energetic metabolism will not offer optimal conditions for virus internalization (which is an active process) and replication. Addressing this point is very relevant because, though missed by the authors, may have influenced the selection of the in vivo therapeutic approach (prophylactic rather than therapeutic) with Phenformin (see comments below).

Indeed the studies mentioned by the reviewer nicely confirm our findings and subtle differences are explainable by the cell systems used. We employed Calu-3 cells, which are considered the most relevant cell culture model, as these are lung cells and exhibit SARS-CoV-2 induced cell death and syncytia formation. Furthermore, we would like to point out that Prade et al, 2023 quote our manuscript as a preprint, that was already released in 2022. We decided to add the compounds 24 hours before infection since the mode-of-action we screened the compounds for was to act as metabolic inhibitors. So we hypothesized that when adding the compounds at a substantial time point after infection, efficacy is likely reduced since it will take some time until the affected pathways are metabolically affected. For the revised manuscript, we added time-of-addition assays (new Figure 7, and new text in l. 402 - 427). These nicely show that there are indeed differential effects of the compounds against the viruses used. For SARS-CoV-2 and RSV there is a TOA effect, whereas DENV and IAV are inhibited in a similar manner, irrespective of the TOA. From this, one could speculate that SARS-CoV-2 as well as RSV are inhibited via differential mechanisms (for instance entry as well as metabolic inhibition) whereas DENV and IAV are only inhibited due to reduced metabolic activity. We included a careful discussion on that, even though we do not want to be too speculative, as confirming this hypothesis would raise the necessity to include another set of comprehensive experiments that go beyond the scope of this study. We believe that it is formally difficult to separate compounds for prophylactic versus therapeutic applications, especially in the field of antiviral drugs, as these compounds are foremost virostatic and inhibit viral spread and infection of as yet non-infected cells. For the compounds presented here, they could be used prophylactically, but they will also have a therapeutic effect in terms of limiting viral spread and replication in a post-infection setting as we now show.

8) If possible, the SARS-CoV-2 infection assay on Calu-3 cells should be repeated and treatment with Phenformin and Atpenin 5 initiated concomitantly with virus infection. This may help to discriminate at which step of the virus cycle these drugs interfere the most.

Thank you for this very important suggestion. We have done these time-of-addition experiments and the results demonstrate that pre-treatment enhances antiviral activity of Phenformin and Atpenin against SARS-CoV-2 and RSV, whereas DENV and IAV do not show a TOA dependent inhibition. This interesting result suggests differential modes-of-action of the two compounds against the viruses employed. Altogether, the phenotypes are compatible with a mode-of-action related to viral attachment, entry and reduced cellular metabolism of Phenformin and Atpenin against SARS-CoV-2 and Atpenin against RSV, whereas DENV and IAV are likely inhibited at the level of metabolic activity, as expected.

These results are now included as new Figure 7:

Figure 7 | Atpenin A5 and phenformin time of addition assay against different RNA viruses. Cells (Calu3, A549 or Huh T7 Lunet RC) were treated with different concentrations of Atpenin A5 or phenformin at several timepoints (24h before infection, 2h before infection, at the same time or 2h after infection). They were infected with either icSARS-CoV-2-mNG (MOI 0.3), respiratory syncytial reporter virus expressing GFP (RSV, MOI 0.3), influenza A reporter virus (IAV, MOI 0.3) expressing GFP or Dengue virus (WT2, MOI 1). Forty-eight hours post-infection, cells were fixed with 2% PFA, stained with Hoechst, and infection rates were measured with a Cytation3 multiplate reader. **A** Cell count (Hoechst+ cells) normalized to mock. **B** Graphs show the infection rate normalized to untreated infected cells. $n=3$; Data represents means \pm S.E.M.

And described in the results (l. 401 - 427) :

“Compound time-of-addition (TOA) analysis suggests differential modes of antiviral activity. In order to get first insights into the potential mode-of-action exerted by the two compounds against the viruses investigated, we performed time-of-addition assays. Compounds were added from 24 hours before infection up to two-hours post infection. Hypothesizing that the compounds suppress metabolism and therefore block viral gene expression, adding the inhibitors two hours after infection, when entry has occurred, should not have a dramatic negative impact on their antiviral activity. Adding atpenin A5 inhibited SARS-CoV-2, RSV, IAV and DENV as expected (Fig. 7). However, of surprise, atpenin A5 showed a clear TOA effect against SARS-CoV-2 and RSV, even though it still exerted some antiviral activity when added 2 hours post infection. This suggests that the compound has a dual mode-of-action against these two viruses, likely involving inhibition of metabolism as well as entry. In contrast, there was clearly no TOA effect of atpenin A5 in suppressing IAV and DENV infection, indicating that these two viruses are inhibited by the compound's action on metabolism. A similar phenotype was observed for phenformin's antiviral activity on SARS-CoV-2 versus DENV (Fig. 7B). In conclusion, while TOA is compatible with the known metabolic suppression of phenformin and atpenin A5, there might be additional antiviral activity exerted by both compounds.”

and the discussion section (l. 522 - 530):

“Furthermore, TOA-assays indicate differential modes-of-action of the compounds used. While DENV and IAV seem mainly inhibited by the compound's effect on metabolism, there is a clear TOA effect of SARS-CoV-2 inhibition by phenformin and atpenin A5 and also RSV is potently suppressed by atpenin A5, especially when it is given before infection. Such a TOA-effect is supportive for a compound mode-of-action related to early steps in the viral cycle, i.e. entry. Of note, SARS-CoV-2 and RSV were also sensitive towards inhibition when the compounds were given after infection, indicating that in addition to the effects on entry, metabolism is also relevant for the observed antiviral activity. ”

9) line 310: please comment on why Scillo-inositol promoted rather than inhibited SARS-CoV-2 replication.

At a very high concentration and with high variability Scillo-inositol seemed to slightly increase infection of SARS-CoV-2 in this pre-screening experiment. This effect was not significant. Furthermore, we have not analyzed this particular phenotype in a follow up experiment as it is out-of-focus and results would not have had an impact on the overall conclusions of our study.

10) lines 328 and 329: The inactivity of phenformin against RSV and IAV should be further discussed in the Discussion section. This result suggests a virus/host-cell-specific action of phenformin and, hence, restricts the emphasis on the "broad spectrum" label assigned to this drug.

Importantly, RSV and IAV were tested in the lung adenocarcinoma cell line A549. While we agree that it is highly interesting that some viruses might be more dependent on cellular metabolic pathways, this effect could be also due to a cell line-specific resistance towards phenformin. This possibility should not be ruled out and the antiviral activity of Phenformin for these two viruses should also be tested in primary cell culture or in vivo, which is beyond the scope of the current study. As suggested, we discuss these aspects in the revised version of the manuscript (l. 630 - 634):

“Furthermore, phenformin was not active in vitro against RSV and IAV. For both infection models we employed A549 cells and these lung adenocarcinoma cells might exert cell-line specific resistance to this compound. So it will be essential to further exploit other cell types and foremost primary cell models, for example air-liquid-interface cultures to verify this phenotype.”

11) The capacity of Phenformin to lower viral load in the upper respiratory tract is promising. However, there are two key factors that, in principle, raise some doubts about its prospective as a "therapeutic" anti-SARS-CoV-2 drug candidate. The therapeutic regimen was initiated 72 hours in advance of infection and failed to control virus replication in a physiologically key target tissue: the lungs. As such, Phenformin (or analogs: Metformin) may be effective as a prophylactic drug transiently applied to risk groups (e.g. health personnel, patients with comorbidities compatible with Metformin or Phenformin administration). The authors may discuss more objectively these points.

We agree that in a clinical setting these drugs would likely be given supportive. As they are different from directly-acting antivirals we would envision an effect that lowers disease severity and improves clinical outcome. In an outbreak situation, these compounds as such could be given to persons-at-risk, as mentioned by the reviewer. However, these scenarios are highly speculative which is why we decided not to discuss them extensively. Instead, we revised and tuned down our statements, accordingly.

12) Concerning the comment above, and taking into account the massive information about the interconnected metabolic pathways relevant for viral replication obtained in this study, it would have been interesting that the authors propose, if feasible, which of them (from Tears-1 targets) may be subject of simultaneous inhibition to achieve synergistic effect.

We thank the reviewer for their insightful comment regarding the potential for synergistic effects through simultaneous inhibition of multiple targets. In our study, however, each target identified independently achieves the desired effect, suggesting that individual modulation is sufficient to address the objective of suppressing viral replication. While exploring combinations could provide further theoretical insights, such investigations would extend beyond the practical scope and priorities of our current work.

13) Please indicate the SARS-CoV-2 variant the virus used for the screening (the mNeon Green...) belongs to.

SARS-CoV-2 mNG is a reporter virus derived from an early Wuhan strain. The reference was added to the manuscript and we now briefly specify this in the M&M section.

Reviewer #2

General Comments

The authors use metabolic network modeling to identify enzymes that, when inhibited, could prevent viral replication by diminishing the rates of energy generation or the production of viral biomass precursors, without adversely affecting the host's cellular metabolism. They include various viruses in their analysis and find common targets, thus making the study relevant for pandemic preparedness. Despite starting with sparse single-cell data, the modeling pipeline successfully highlights differences in viral replication potential between cells from infected and healthy individuals, and predicts targets that are known to interact with viral proteomes. It is particularly interesting and insightful that cells that are from patients but are not yet infected are predicted to be metabolically primed to support viral replication. The authors then focus on a subset of target proteins that have been experimentally verified to interact with these viruses. Two available drugs that target proteins from this group are experimentally validated to have significant potential. It is not feasible to validate systems-level modeling predictions in a high throughput manner. Also, since the choice of targets and drugs for validation relied on their previously known interactions and historical data on their relevance and success potential, it is difficult to fully assess this modeling pipeline's utility for therapeutic applications. Therefore, it is critical that the modeling approach is free of concerns (please see my major comments) and delivered clearly (please see my minor comments).

Major Comments

In principle, halting the replication of a virus (or the proliferation of a tumor) by suppressing biomass or energy production is straightforward, but maintaining patient health during this process is a significant challenge. The modeling predictions of this study are based on the biomass reactions of both the virus and the human host. Thus, a critical part of the analysis involves optimizing potential targets, defining an ideal target as one that maximally reduces viral biomass production while minimally affecting human biomass production. The Methods section specifies two thresholds for defining these targets: a maximum 50% reduction in viral biomass and a minimal 20% reduction in human biomass (I am omitting the details on tier-1 and tier-2 targets for simplicity). Despite the importance of this optimization, the rationale for choosing these particular thresholds and the workings of this strategy are unclear, so I have several questions:

1) Given the shared components and ATP requirements in the biomass reactions of both the virus and humans, what makes viral replication so vulnerable when human biomass production is not?

We thank the reviewer for pointing this out. However, please note that we specifically search for targets that impede viral replication while having a negligible effect on biomass production. Thus, our results do not imply that viral replication is more vulnerable to interventions than cellular growth. Also there are considerable differences in the molecular composition of viruses versus human cells. For instance, the viruses we considered do not contain any DNA nor do they require vitamins or minerals etc. From a biological perspective, viral replication typically entails a massive induction of metabolism as we also observe with

almost all pathways being upregulated (e.g. Fig. 1C) required for the rapid synthesis of building blocks of virions. This might make virally infected cells potentially much more susceptible to inhibition of metabolism compared to normal (non-dividing) human cells who invest most of their energy budget into maintenance of cellular function that can also include salvage which is typically metabolically much cheaper than de novo synthesis of cellular building blocks.

2) Why do enzymes involved in the TCA cycle and electron transport chain appear non-essential in the human metabolic model, not contributing to significant reductions (>>20%) in human biomass production?

If an enzyme is selected as a tier-1 target that does not mean that it is not essential for cellular growth in all cell types. For instance, using the BALF-2 data set from SARS-CoV-2 infected patients as reference, we find that among the ~140k sequenced cells, enzymes of complex I of the respiratory chain were identified in 12.8% of the cells as tier-1 target and in 3.6% of the cells as tier-2 target (i.e. also essential for biomass formation). Thus, complex I is also essential for maximal biomass production in some cells but viral replication appears to be much more dependent on it. Also, from a purely modeling perspective, it is possible that biomass production is limited by some of the compounds that are not required for viral replication.

3) How sensitive is the outcome to the thresholds used? Could these thresholds be overfitted to known druggable targets in the TCA cycle and ETC?

We thank the reviewer for this valuable comment. Indeed, we have selected those thresholds empirically. Unfortunately, due to the number of FBA runs that needed to be performed for this analysis (270k cells x 1681 genes x two tests, equaling ~ 800 million data points), we only stored results for genes if they reduced viral replication by at least 50% and cellular growth to no less than 80%. Thus, we can only perform a sensitivity analysis within these bounds (i.e. maximal viral biomass reduction between 0 to 50% and maximal biomass production rate between 80 to 100%). For changes in the cutoffs for the biomass production rate, we found that only CDIPT was sensitive to variations in that cut-off and was not identified as a tier-1 target for 90% or 100% cut-offs on changes in biomass production rate upon knockout. Concerning variations in required maximal viral replication capacity, we found that SDHA and SLC16A1 were identified as tier-1 targets for viral replication for cutoffs down to 30% and complex I for cutoffs down to 20%. Thus, the targets we have successfully validated are relatively insensitive to our parameter choice while CDIPT, for which we could not confirm an antiviral effect of the corresponding tested compound in vivo, was sensitive to the chosen biomass cutoff.

We added the following description and a new panel (Fig. 3C) describing the sensitivity analysis (l. 252 - 257):

“We evaluated the sensitivity of our selected targets to variations in the cut-off thresholds for maximal viral replication rate and maximal biomass production rate. Our analysis revealed that targets associated with complex I, complex II and SLC16A1 were largely insensitive to changes in these cut-offs. In contrast, CDIPT was identified only at cut-off values of 80% or higher for maximal biomass production rate, but not at more stringent thresholds (Fig. 3C).”

Fig. 3C Sensitivity analysis of identified broad-spectrum antiviral targets. Cut-offs used for the analysis are indicated in the plots. Black rectangles represent genes that are identified as broad-spectrum antiviral targets in the corresponding setting. Bold gene names correspond to those selected for experimental validation.

4) How do the predicted common targets compare with straightforward predictions from traditional flux balance analysis using the biomass objective functions without integrating any data?

We thank the reviewer for this insightful comment. Applying the used cutoffs (<50% maximal viral replication rate, >80% maximal biomass production rate) on the human metabolic model with our predefined diet without mapping of transcriptomic data, we do not identify any tier-1 targets. Actually, there are only two knockout targets that reduce viral replication rate (asparagine synthetase (ASNS) and cytidine/uridine monophosphate kinase 1 (CMPK1)) but ASNS knockout reduces maximal biomass production rate to 57% while CMPK1 knockout is lethal. However, we have not added this analysis to the main manuscript since we feel that it might be more distracting than providing insights into our choice of analysis.

For this work to be convincing, these aspects must be thoroughly addressed, possibly requiring additional analyses and figures.

Minor Comments

5) Uptake rates (Table S8): At a first glance, it makes no sense to directly use metabolite concentrations (units of mmol/L) as exchange flux rates (units of mmol/g.h). But, I understand that this is considered as a first approximation that is acceptable in a semi-quantitative modeling setting (FASTSCORE integration). And I agree that it may be sufficient for this case. But this point deserves some clarification and more discussion for the modeling audience.

We thank the reviewer for this comment. While we agree that uptake rates are certainly distinct from the concentration of a compound, this is a simplification step that is typically

done when working with complex growth media (like e.g. the blood). E.g. in microbial community modeling, diets are translated into the molar concentrations of the corresponding compounds and then used as input to the metabolic models. (see e.g. <https://doi.org/10.1038/nbt.3703> and <https://doi.org/10.1371/journal.pone.0236890>). We have added the following statement to the methods section (l. 683 - 685):

“Please note that this diet approximates maximal uptake rates for metabolites based on their concentration, as typically done when considering complex media⁹².”

6) Lines 653-655: "Gene level counts were translated into transcripts per million (TPM) values through normalization according to human ENSEMBL gene lengths". Single cell data is based typically on 3' or 5' end reads, not full-length reads, and therefore gene length is irrelevant in most cases. I checked only one of the datasets that the authors used (Liao et al., 2020; data can be traced back to <https://www.ncbi.nlm.nih.gov/geo/query/acc.cgi?acc=GSM4339769>), and got the impression that it is based on 5'-end reads. Please clarify this matter regarding data normalization.

We thank the reviewer for this comment. Indeed we performed TPM normalization because it is a de facto standard for bulk sequencing data and there was not yet a consensus on whether TPM normalization is appropriate for scRNA-Seq data when we initiated this project in 2020. In principle, a log normalization approach accounting for read counts per cell is the more appropriate but would require us to re-run our entire pipeline. Thus, we have opted to keep this normalization method, but now acknowledge in the methods that other normalization approaches are more suited for this type of scRNA-Seq data (l. 724 - 727):

“Please note that while we used TPMs, the protocols used for scRNA-Seq often involve sequencing from the start or end of a transcript and thus approaches for normalization not involving gene length might be more suitable for this type of data.”

7) Discussion: A big part of the discussion is repeating the results (including but not limited to the second and third paragraph). Please consider revising to focus more sharply on critical topics needing further exploration and clarification.

We thank the reviewer for this comment and have considerably streamlined the discussion to reduce redundancies with the results section.

8) Fig. 2, legend: The opening is unclear and convoluted. Please rephrase.

We have rephrased the opening of the figure legend to: “Enrichment of viral interaction partners among model-predicted enzymes involved in the replication of SARS-CoV-2, dengue virus, and influenza A virus. The fraction of experimentally identified viral proteome interaction partners among tier-1 and tier-2 targets (found in at least 5% of cells with tier-1 targets) was calculated for each dataset.”

9) Line 311: I believe the referral to Fig 4B is a typo and it should be 4C.

We thank the reviewer for spotting this typo. Indeed, we wanted to refer to Fig. 4C.

10) Line 329: The manuscript claims no severe cytotoxicity was observed for the compounds tested, yet some data points show viable cell counts below 50%. Is this level not considered toxic?

We are not sure to which data points the reviewer is referring to. However, it is quite normal and expected that compounds exhibit toxicity at high concentrations. This is essential to calculate the CC50 and allows us to determine the therapeutic index or selectivity index (SI), which basically indicates how specific a compound exerts a certain effect (antiviral activity) over general toxicity. For our compounds these values are summarized in table 1. Usually a value >100 is considered a superior SI.

11) Fig. 4, legend: mNG is used in abbreviated and non-abbreviated form in different places

We unified this and introduced mNG once, using the abbreviation then afterwards throughout

12) Line 421: What is ARN? (Possibly a typo for RNA, but this is repeated within the same paragraph as well as in Fig 7)

We thank the reviewer for spotting this typo. Indeed, we meant RNA.

12) Line 463: The assertion that virus-induced cellular senescence contributes to a pro-inflammatory microenvironment is counterintuitive if senescence is supposed to coincide with increased metabolic activity necessary for viral replication. This paradox needs clarification.

We agree with the reviewer that our statement that cellular senescence could contribute to the observed pleiotropic effect of SARS-CoV-2 infection is counterintuitive since cellular senescence is typically associated with reduced metabolic activity. Thus, we have removed this statement from the manuscript.

13) Line 599, Table S6: This reference should likely be to "Table S7." Additionally, the terms "VBOF" are used to refer both to viral biomass objective functions and viral replication reactions, which could confuse readers. Please eliminate the latter term to avoid confusion.

The reference to Supplementary Table S6 is correct as it contains the protein stoichiometry of the viruses considered. We replaced the term VBOF by "viral replication reaction" throughout the manuscript.

Requests from the editor:

-Please repeat the infection assay as suggested by R1, with treatment with inhibitors at the same time as infection start.

Please see our related response to reviewer 1, comment #7.

-Please carry out more experiments to explore how sensitive the outcome is to the thresholds used, and how the predicted common targets compare with predictions from flux balance analysis without integrating any data.

Please see our related response to reviewer 2, comments #3 and #4.

Reviewer #1 (Remarks to the Author):

(our responses in italics in blue, changes pertaining to specific reviewer comments are also indicated in the version of the manuscript containing tracked changes, line numbers refer to the document containing tracked changes)

The authors considered the most relevant suggestions (including new experimental information) made in the revision and satisfactorily answered all the questions raised. I have no more concerns and recommend its publication.

We thank you for this very positive assessment of our work!

Reviewer #2 (Remarks to the Author):

1) The authors have properly addressed my major comment on sensitivity and all minor comments. However, concerns about the source of mechanistic predictions (specifically, how the model predicts that the perturbation of an enzyme significantly impacts viral replication a lot more than human biomass production) remain inadequately addressed. The responses provided in the rebuttal are vague, and no corresponding modifications have been made to the manuscript for readers to evaluate these claims.

Please see our detailed responses below.

2) I also see that some of the other comments could not be perfectly addressed because redoing the computational analysis is challenging. This difficulty underscores the importance of clearly elucidating the inner workings of the predictive algorithms. If executing this modeling task is challenging even for the authors, who have the pipeline readily available, how can a potential user of the approach, like myself, confidently launch a project without being convinced of its robustness?

We assume the reviewer is referring to our statement that we could only perform the sensitivity analysis on a subset of possible parameter values since we have not stored all results of the FBA simulations. The question of repeating these simulations is rather one of practicality rather than complexity. After all, these are simple standard FBA simulations of the models we already reconstructed and provided along with this manuscript. However, as we mentioned in our previous response, this would require running 800 million FBA simulations (approx. 2400 CPU hours). This computation can undoubtedly be accomplished, but we are not convinced that any outcome of repeating those simulations would justify the amount of computational resources and carbon footprint this would entail. With our sensitivity analysis, we have checked whether our predictions hold up to more conservative parameter settings (e.g., lower required maximal viral replication rate or higher maximal biomass), so we could only test the sensitivity to more relaxed parameter settings regardless.

3) As I mentioned previously, validating only two predictions out of hundreds does not sufficiently demonstrate the utility of the model, especially when those targets were selected based on other sources of evidence as well. The only way to build confidence in a model of

this scale is through internal validation, which would explain the reasoning behind its predictions.

We disagree with this statement. As we describe in the manuscript, we tested four targets, and since they are part of enzyme complexes, they cover 11 out of 39 predicted targets. Modeling alone indeed predicted 254 shared tier-1 targets. However, we do not see the point in not considering further prior knowledge to subselect targets, which we did by incorporating information about experimentally determined host-viral interactions. After all, we do not claim that we obtain high-confidence targets by modeling alone, but the subselection based on systematic prior knowledge is part of our pipeline. Also, running those validation experiments is highly laborious, as seen from the extent of the downstream experimental analysis we describe in our manuscript. Thus, testing further targets would certainly be possible but would go well beyond the scope of our manuscript.

Indeed, as we already report in our manuscript only considering the modeling we see a highly significant enrichment of known viral interaction partners among purely model-predicted targets (l. 215 - 219):

“Using the BioGRID database ³⁴ as a reference, we find that interactions of 158 of these targets [out of 254] with human pathogenic viruses have been reported before, which represents a highly significant enrichment among reported interactions between enzymes and viruses (Fisher’s exact test p -value = 9.1×10^{-12} , odds ratio 1.9-3.4) and supports their central role in viral replication across diverse viruses.”

Finally, let us remove the modeling and apply only our remaining selection criteria (i.e., interaction with at least two viral proteins). We obtained 84 enzyme targets and thus a much higher number of candidates than the 39 obtained by combining modeling with these selection criteria. We now also state this in the manuscript (l. 252 - 253):

“Please note that using protein interactions as selection criterion alone would yield 84 targets (Supplementary Table S2).”

We acknowledge that one might argue our modeling pipeline's application, reducing 84 to 39 targets, may seem not worth the effort. However, this must be evaluated in the context of a pronounced selection for viral interactions with enzymes critical for viral replication. Among the 1691 genes in the human model, 254 (15%) were identified as tier-1 targets, and of these, 39 (46%) are strong interaction partners for the three viruses studied. This represents a highly significant enrichment compared to the 84 (5%) of all enzymes that are strong interaction partners across the entire human metabolic model.

4) Since some of my comments appear to have been misunderstood or misinterpreted, I will now explicitly outline what a satisfactory response would look like, offering an example to avoid further ambiguity.

To clarify the source of modeling predictions, I would suggest the following:

*Take two example predictions, possibly from Fig. 3B (a blue and a red reaction). Show that regular FBA using a generic model would fail to predict these outcomes (if it doesn't, an alternative example can be chosen to replace one of these).

As outlined in our previous response, running a standard FBA optimizing for viral replication rate on the generic human model with the corresponding medium but without contextualization (i.e., integration of gene expression data) does not yield any target with the cutoffs we defined. Quoting from our previous response: "Applying the used cutoffs (<50% maximal viral replication rate, >80% maximal biomass production rate) on the human metabolic model with our predefined diet without the mapping of transcriptomic data, we do not identify any tier-1 targets. There are only two knockout targets that reduce viral replication rate (asparagine synthetase (ASNS) and cytidine/uridine monophosphate kinase 1 (CMPK1)), but ASNS knockout reduces maximal biomass production rate to 57%, while CMPK1 knockout is lethal. However, we have not added this analysis to the main manuscript since we feel that it might be more distracting than providing insights into our choice of analysis."

5) *Demonstrate that something specific to the integrated models leads to these predictions. For instance, it might be that the TCA cycle is relatively less important in the target cells, as indicated to the integration algorithm via gene expression data.

See our response to your comment #3.

6) *Next, explain how the energy requirements of these cells are met (e.g., through glycolysis, beta oxidation, or possibly reduced overall energy demands as predicted by the integration).

*Ideally (not necessarily), use this analysis to connect the modeling predictions to experimental results, such as those summarized in Fig. 6.

We do not see the point of the reviewer. We can only speculate how these cells meet their energy requirements, which is beyond our analysis's scope. As outlined in our previous response, viral replication seems to depend more on the respiratory chain than biomass production. Also, that cells treated with inhibitors of the respiratory chain are viable is confirmed by toxicity assays we have performed, in vivo data in Syrian hamsters that we provide, and, in the case of phenformin, even by clinical studies in humans. Also, the strong dependence of viral replication on cellular respiration has been documented before^{1,2}. To alleviate some of the reviewer's concerns, we now discuss the role of the respiratory chain in more detail in the results section, mentioning points from our previous review response.

Additionally, we have used the original biomass reaction and the VBOF reaction that we have included in the model and compared their relative energy requirements. Based on the stoichiometry of the corresponding reactions, we found that forming 1 gram of biomass requires 5.9 mmol of ATP while forming 1 gram of virions requires 19.6 mmol of ATP. Thus, in line with prior experimental evidence, virion production is energetically much more demanding than biomass formation. We have added the following text to the manuscript (l. 229 - 239):

“It is interesting to note that the knockout of the respiratory chain is predicted to mostly affect viral replication but to a lesser extent normal biomass production. Exploring this observation in more detail using the BALF-2 data set (data from patients with COVID-19), we find that among the ~140k sequenced cells, enzymes of complex I of the respiratory chain were identified in 12.8% of the cells as tier-1 target and in 3.6% of the cells as tier-2 target. Thus, complex I is also essential for maximal biomass production in some cells but viral replication appears to be much more dependent on it. This is also reflected in a comparison of the ATP maintenance costs of the normal biomass reactions of the human model and the viral replication reaction. While the production of one gram of cellular biomass requires 5.9 mmol of ATP, the production of 1 gram of virions requires 19.6 mmol of ATP, supporting a much higher energy demand of viral replication.”

and in the methods (l. 772 - 779):

“To determine the energetic requirement of normal growth versus viral replication, we first determined the total weight of compounds required for a set flux of 1 mmol/gDW/hr in the biomass reaction and the viral replication reaction. To this end, the corresponding stoichiometric coefficients of each compound consumed for biomass or virion production were multiplied with the molecular weight of the corresponding compound. The amount of ATP hydrolyzed to ADP as part of the biomass or viral replication reaction (29.25 mmol/gDW/hr for biomass, 30.62 mmol/gDW/hr for viral replication) was subsequently divided by this weight to obtain the amount of ATP required for production of one gram of biomass or virions.”

I must stress that this is just a hypothetical example to clarify how the missing elements could be addressed; the actual implementation will depend on the data and findings at hand. For example, the example chosen might be about a biomass precursor that the virus needs but the human cell does not. Nonetheless, this level of detail is essential to understand and trust the mechanistic basis of the predictions. While showing subsystem enrichments of predictions is a helpful start (Fig. 3A), it does little to explain the mechanisms driving those predictions.

References

1. Moreno-Altamirano MMB, Kolstoe SE, Sánchez-García FJ. Virus Control of Cell Metabolism for Replication and Evasion of Host Immune Responses. *Front Cell Infect Microbiol.* 2019;9. doi:10.3389/fcimb.2019.00095
2. Mahmoudabadi G, Milo R, Phillips R. Energetic cost of building a virus. *Proc Natl Acad Sci U S A.* 2017;114(22):E4324-E4333. doi:10.1073/pnas.1701670114

Reviewer #1:

1) Their arguments on point 3 are consistent with the parametrization and selection filter for proposing the candidate host genes contributing to viral replication. Eventually, to address the reviewer's concern, they may mention any potential caveat of the search algorithm strategy used to notify other researchers about some limitations the method may have or to open new avenues for improvement.

We thank the reviewer for this suggestion. We have expanded the limitations section of the discussion to discuss potential biases introduced by subsetting targets with protein interaction data (l. 507 - 511):

“Our subsetting approach might further focus predicted targets to those with known interactions with viral proteins since experimental approaches for detecting such interactions are likely biased to more abundant proteins. Additionally, by concentrating on targets that can inhibit replication across several viruses, we might have missed potent targets for viral inhibition in individual viruses.”

2) Regarding point 4, the authors have provided direct evidence that, if not combined with transcriptomic data, the FBA prediction model fails to pick up physiologically relevant candidate genes. Although they prefer not to comment on the results of these control

experiment, to attend the pertinent request of the reviewer, they can briefly mention in the manuscript (Mat&Met or Result section, when discussing how the search strategy was validated) something like this: "Applying the used cutoffs (<50% maximal viral replication rate, >80% maximal biomass production rate) on the human metabolic model with our predefined diet without the mapping of transcriptomic data, did not yield any tier-1 targets. From the two knockout targets identified as candidates to affect viral replication (asparagine synthetase: ASNS, and cytidine/uridine monophosphate kinase 1: CMPK1), ASNS knockout reduces maximal biomass production rate to 57%, while CMPK1 knockout is lethal. This result highlights the relevance of combining generic FBA modeling with gene expression data."

We have accordingly added a slightly reworded statement in the methods section (approx. l. 668 - 674):

"Please note that applying the used cutoffs (<50% maximal viral replication rate, >80% maximal biomass production rate) on the human metabolic model with our predefined diet without the mapping of transcriptomic data did not yield any tier-1 targets. However, two tier-2 targets could be identified: asparagine synthetase (ASNS) and cytidine/uridine monophosphate kinase 1 (CMPK1). ASNS knockout reduces maximal biomass production rate to 57%, while CMPK1 knockout is lethal. This result highlights the relevance of combining the generic model of viral replication with gene expression data."

3) Concerning point 6, I agree with the authors that it is not necessary to explain the cells' energy requirements, which, independent of viral infection, will depend on nutrient/substrate availability, cell cycle stage, and other less important factors. Clearly, viral replication demands a lot of ATP and the most efficient energetic metabolism to produce it is the TCA-cycle coupled to the mitochondrial electron-transport chain. To support this statement, the authors should consider that nucleic acid biosynthesis is an energetically costly process; at least one mol ATP is consumed for each base ligated to the nucleic acid chain (RNA in the case of SARS-CoV2). Therefore, any pharmacological strategy reducing cellular ATP will efficiently affect the replication of non-integrative and highly replicating viruses.

We thank the reviewer for this suggestion. This cost is already included in the viral replication reaction, which also consumes ATP as part of the production of virions (as we state in the methods section, l. 547 - 549). Thus, our model uses ATP as an energy carrier and a molecular building block.